# FlexPlanner: Flexible 3D Floorplanning via Deep Reinforcement Learning in Hybrid Action Space with Multi-Modality Representation

**Ruizhe Zhong[1], Xingbo Du[1], Shixiong Kai[2], Zhentao Tang[2], Siyuan Xu[2],**
**Jianye Hao[2,3], Mingxuan Yuan[2], Junchi Yan[1]**[*]
[1]Dept. of CSE & School of AI & MoE Key Lab of AI, Shanghai Jiao Tong University
[2]Noah's Ark Lab, Huawei
[3]College of Intelligence and Computing, Tianjin University
{zerzerzerz271828, duxingbo, yanjunchi}@sjtu.edu.cn
{kaishixiong, tangzhentao1, xusiyuan520, yuan.mingxuan}@huawei.com
jianye.hao@tju.edu.cn

## Abstract

In the Integrated Circuit (IC) design flow, floorplanning (FP) determines the position and shape of each block. Serving as a prototype for downstream tasks, it is critical and establishes the upper bound of the final PPA (Power, Performance, Area). However, with the emergence of 3D IC with stacked layers, existing methods are not flexible enough to handle the versatile constraints. Besides, they typically face difficulties in aligning the cross-die modules in 3D ICs due to their heuristic representations, which could potentially result in severe data transfer failures. To address these issues, we propose FlexPlanner, a flexible learning-based method in hybrid action space with multi-modality representation to simultaneously handle position, aspect ratio, and alignment of blocks. To our best knowledge, FlexPlanner is the first learning-based approach to discard heuristic-based search in the 3D FP task. Thus, the solution space is not limited by the heuristic floorplanning representation, allowing for significant improvements in both wirelength and alignment scores. Specifically, FlexPlanner models 3D FP based on multi-modalities, including vision, graph, and sequence. To address the non-trivial heuristic-dependent issue, we design a sophisticated policy network with hybrid action space and asynchronous layer decision mechanism that allow for determining the versatile properties of each block. Experiments on public benchmarks MCNC and GSRC show the effectiveness. We significantly improve the alignment score from 0.474 to 0.940 and achieve an average reduction of 16% in wirelength. Moreover, our method also demonstrates zero-shot transferability on unseen circuits. Code is publicly available at: `https://github.com/Thinklab-SJTU/EDA-AI`.

## 1 Introduction

In the very beginning stage of physical design in Electronic Design Automation (EDA), floorplanning (FP) plays a critical role. As a subsequent stage of hardware design [1] and logic synthesis [2], floorplanning provides a prototype for downstream tasks [3], ranging from power delivery network (PDN) design [4] to placement [5, 6] & routing [7, 8] (P&R), hence determining the upper bound of final PPA (Power, Performance, Area). Recognized as an NP-hard problem [9], FP establishes

---

[*]Corresponding Author. This work was partly supported by NSFC (62222607, 92370201) and Shanghai Municipal Science and Technology Major Project under Grant 2021SHZDZX0102.

the chip's physical layout by optimizing the position and shape of the major blocks to minimize interconnect lengths and ensure efficient silicon area utilization. As floorplanning technologies evolve, 3D FP with stacked layers emerges with more challenges. In particular, the cross-die module alignment becomes another pivotal factor in 3D FP [10, 11, 12]. For instance, vertical buses [11] for cross-die communication connect the aligned blocks spread among multiple dies [11]. Another example is the Memory-on-Logic technology [12, 13, 14], partitioning the processor [15] into two tiers: memory tier and logic tier. The memory tier consists of memory blocks and customized intellectual property cores (IP cores) [16], while the logic tier contains other components, such as logic blocks. Blocks on different dies should be aligned together, enabling the communication established by bonding bumps or pads [14].

Existing works can be categorized into heuristic-based methods, analytical approaches, and learning-based methods. Heuristics-based methods [17, 18, 19, 20, 21] model the FP with a certain heuristic representation. To refine the current FP, they modify the heuristic representation and convert it to the corresponding FP through a decoding scheme. However, this implementation limits the flexibility to directly adjust the position of blocks. After a single modification, the entire FP result needs to be regenerated, incapable of making fine-grained adjustments. Moreover, the alignment constraints cannot be satisfied by simply incorporating alignment metrics into its heuristics. Analytical approaches [9, 22] compute the gradient of objectives w.r.t. block position and utilize gradient descent technique to optimize FP. However, the calculation of alignment is non-differentiable, making them inapplicable in 3D FP. Recently, with the emergence of Reinforcement Learning (RL), learning-based methods are promising in FP [23, 24, 25, 26, 27, 28, 29]. Among these approaches, [23, 24, 25] still retain traditional FP representation, thereby leading to the same issues encountered by heuristic methods. Conversely, [26, 27, 28, 29] focus on determining block positions in 2D scenarios. However, when directly implemented in 3D scenarios, these methods could result in 1) overlooking alignment requirements and 2) multi-die property. Specifically, in 2D FP, blocks are arranged exclusively on a single die and organized in a queue that represents the placing order of all blocks. However, in 3D scenarios, multiple queues exist due to the multi-die property, necessitating a layer decision mechanism to merge these queues and determine the comprehensive placing order. This is a critical issue, yet it remains insufficiently explored in current research. Additionally, current learning-based methods are 3) incapable of addressing the variable aspect ratio of modules. Characteristics of typical approaches are summarized in Table 1.

To address the aforementioned challenges, we propose **FlexPlanner**, a flexible deep-learning-based approach in hybrid action space with multi-modality representation for 3D FP. Flex-Planner directly outputs the final FP result, without the reliance on any heuristic representation. Under the Actor-Critic framework, the policy network consists of three sub-modules, responsible for determining the position, layer, and aspect ratio of blocks, spanning a hybrid action space. Empirical results demonstrate the effectiveness and significance of FlexPlanner. **The main contributions are highlighted as follows:**

Table 1: Characteristics of typical methods.

| Method | Type | AR | Aln | 3D | Mod |
|---|---|---|---|---|---|
| PeF [9] | Analytical | ✗ | ✗ | ✗ | N/A |
| 3D-B*-SA [17, 21] | Heuristics | ✗ | ✗ | ✓ | H |
| Wiremask-BBO [30] | Heuristics | ✗ | ✗ | ✗ | V |
| RL-CBL [24] | Heuristics, RL | ✗ | ✗ | ✗ | H, G |
| GraphPlace [26] | RL | ✗ | ✗ | ✗ | G |
| DeepPlace [27] | RL | ✗ | ✗ | ✗ | V, G |
| MaskPlace [28] | RL | ✗ | ✗ | ✗ | V |
| Ours | RL | ✓ | ✓ | ✓ | V, G, S |

AR: aspect ratio of blocks. Aln: cross-die block alignment. Mod: Modality.
*H = Heuristics, V = Vision, G = Graph, S = Sequence.

- **First learning-based method to discard heuristic-based search in the 3D FP task.** We propose a novel learning-based method with flexible hybrid action space for 3D FP, simultaneously handling the position, aspect ratio, and cross-die alignment of blocks. Without relying on the heuristic-based search, FlexPlanner allows the position and aspect ratio of each block to be explored across a comprehensive spectrum, rather than be limited by the constraints of heuristic FP representation, thereby breaking through the upper bound of performance. And we propose an innovative strategy for more effectively addressing the alignment issue.

- **Tackle the non-trivial issue of dependency on heuristics by incorporating hybrid action space and multi-modality representation.** It is non-trivial to avoid the dependency on heuristics-based search in 3D FP due to the difficulty of modeling the complex solution space. Heuristics can only represent a subset of the entire solution space, resulting in limitations on the upper bound performance. To address this issue, we initially introduce three modalities, including vision, graph, and sequence, to comprehensively represent the state space. Additionally, we design a sophisticated policy network with hybrid action space and asynchronous layer decision mechanisms, enabling learning versatile properties such as position, aspect ratio, layer for each block in a 3D FP setting.

- **Zero-shot transferability.** Leveraging the advantage of the learning scheme and multi-modalities, FlexPlanner demonstrates the ability to exhibit zero-shot transferability on previously unseen circuits. This capability shows strengths in its efficiency, as it conserves substantial training resources when confronted with new IC cases.
- **SOTA experimental results with significant alignment improvement.** Within the learning framework, FlexPlanner achieves state-of-the-art performance on wirelength and alignment in 3D FP. Specifically, the average reduction in wirelength arrives at 16%, compared to previous works. Moreover, by effectively incorporating the alignment constraint, we achieve 0.940 on the alignment score, significantly surpassing the previous SOTA score of 0.474.

## 2 Preliminary and Formulation

**Floorplan.** The 2D floorplan task aims to determine the position and shape of each block given the block list, I/O port list, and netlist. Based on this, the 3D floorplan task is further required to place all blocks across multiple dies/layers. Each *die/layer* $d \in \mathcal{D}$ is a rectangular region with width $W$ and height $H$, and all dies are of the same shape. Specifically, the block list is denoted as $\mathcal{B} = \{b_1, b_2, \ldots, b_n\}$ with $n$ blocks, where each *block* $b_i$ is a rectangle with width $w_i$, height $h_i$, and area $a_i = w_i \cdot h_i$. The bottom-left coordinate of block $b_i$ is denoted as $(x_i, y_i)$, and the layer where the $b_i$ is located is denoted as $z_i$. There are two types of blocks: *hard block* and *soft block*. For a *hard block*, its aspect ratio $\text{AR}_i = \frac{w_i}{h_i}$ is fixed. For a *soft block*, $\text{AR}_i$ can vary between $[\text{AR}_{\min}, \text{AR}_{\max}]$, while the area must always satisfy $a_i = w_i \cdot h_i$. Additionally, we denote the I/O port list as $\mathcal{T} = \{t_1, t_2, \ldots, t_m\}$ with $m$ *ports* [2]. Each *port* $t_j$ is viewed as a point with a pre-determined position $(x_j, y_j, z_j)$, where $(x_j, y_j)$ is the coordinate, and $z_j$ is the layer index. *Netlist* is defined as a set consisting of all nets, where each *net* is a set of blocks and ports, representing the interconnections. Since the layer $z_i$ of block $b_i$ is pre-assigned, we only need to determine the coordinate $(x_i, y_i)$ and the aspect ratio $\text{AR}_i$ for all blocks to optimize the following objectives:

**1) Alignment.** Given two blocks $b_i, b_j$ on different layers, *alignment* evaluates the overlap/intersection area between them on the common projected 2D plane. We define the alignment score $\text{aln}(i, j)$:

$$
\begin{aligned}
\text{aln}_x(i, j) &= \max\left(0, \min(x_i + w_i, x_j + w_j) - \max(x_i, x_j)\right), \\
\text{aln}_y(i, j) &= \max\left(0, \min(y_i + h_i, y_j + h_j) - \max(y_i, y_j)\right), \\
\text{aln}(i, j) &= \min\left(1, \frac{\text{aln}_x(i, j) \cdot \text{aln}_y(i, j)}{\text{aln}_m(i, j)}\right),
\end{aligned}
\tag{1}
$$

where $\text{aln}_m(i, j)$ is the required minimum alignment area between block $b_i$ and $b_j$. The total alignment score should be ***maximized*** to satisfy the alignment requirement. $(b_i, b_j)$ forms an alignment pair if $\text{aln}_m(i, j) > 0$, and $b_i, b_j$ are mutual *alignment partners*.

Cross-die block alignment is common in 3D FP [11, 13, 14]. Taking two blocks in different dies as an example, vertical buses [11] or bonding bumps/pads [14] are employed for communication, which requires capabilities for cross-die block alignment. That is, considering their projection onto a 2D plane, the related blocks must exhibit some minimum intersecting region, denoted as $\text{aln}_m(i, j)$.

**2) HPWL and 3) Overlap.** Half Perimeter Wire Length (HPWL) is an approximate metric of wirelength. It can be computed much more efficiently, as accurate wirelength can be accessed only after the time-consuming routing stage. The summation of HPWL should be ***minimized***:

$$
\sum_{\text{net} \in \text{netlist}} \left( \max_{m_i \in \text{net}} x_i^c - \min_{m_i \in \text{net}} x_i^c + \max_{m_i \in \text{net}} y_i^c - \min_{m_i \in \text{net}} y_i^c \right),
\tag{2}
$$

where $m_i$ is either a block or port in net and $x_i^c$ is the center x-coordinate. For a block, $x_i^c = x_i + \frac{w_i}{2}$, and for a port, $x_i^c = x_i$. Given two blocks $b_i, b_j$ on the same die, the overlap area between them should be ***minimized*** (detailed calculation is given in Alg. 4 in Appendix G.1). Besides, all blocks should be placed within the fixed outline.

---

[2]It is also commonly referred to as terminal.

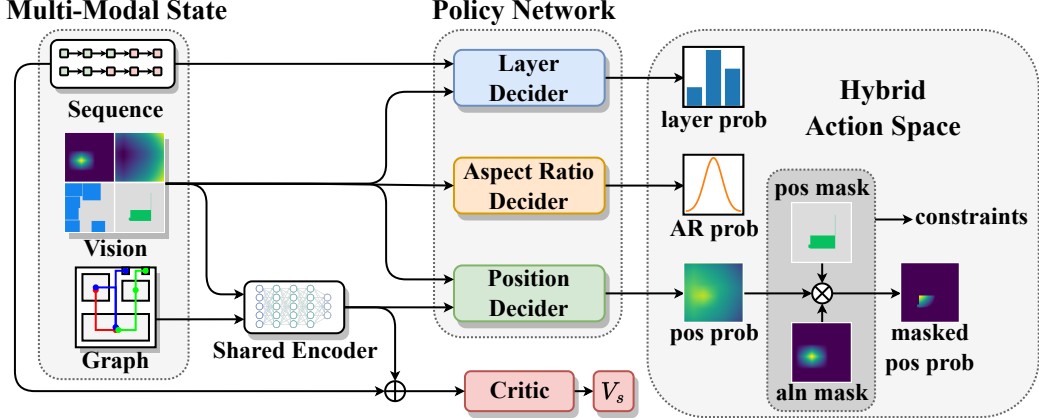

Figure 1: Pipeline of FlexPlanner. Under the Actor-Critic framework, taking the multi-modality representation as input, the policy network consists of three sub-modules, responsible for determining the position, layer, and aspect ratio of blocks. Alignment mask and position mask are incorporated to filter out invalid positions where constraints (alignment, non-overlap, etc.) are not satisfied.

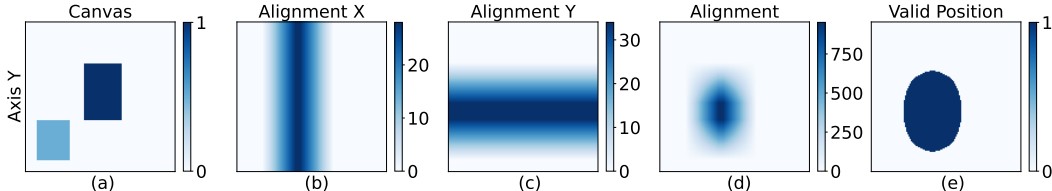

Figure 2: Demonstration of alignment. In (a), the light blue region will be occupied by the block to place, and the dark blue region is occupied by its alignment partner block which has been placed. By sliding block to place across the plane, we obtain the alignment values at each position along the X and Y dimensions, as shown in (b) and (c). Final alignment can then be calculated through element-wise matrix multiplication, as illustrated in (d). Only $(x, y)$ satisfying $\text{aln}_x \cdot \text{aln}_y \geq \text{aln}_m$ are valid positions shown in (e), and this binary mask can be incorporated to filter out invalid positions.

## 3 Methodology

**Overview.** The 3D floorplanning task can be formulated as an episodic Markov Decision Process. As shown in Fig. 1, pipeline of our approach mainly includes state, hybrid action space, policy network and critic network. State $s_t$ consists of three modalities, including vision, graph, and sequence. Action $a_t$ is represented as $(x, y, z, \text{AR})$, where position $(x, y, z)$ are discrete variables and aspect ratio AR is a continuous variable. These properties form a hybrid action space. Under the Actor-Critic framework, critic $V_\phi(s_t)$ evaluates the current state. Policy network $\pi_\theta(a_t|s_t)$ determines the 2D coordinate $(x_t, y_t)$ of current block $b_t$, the layer $z_{t+1}$ to access next block $b_{t+1}$, and the aspect ratio $\text{AR}_{t+1}$ of $b_{t+1}$. To explicitly impose constraints on the action space, masks are applied to the probability matrix of block positions. This process filters out invalid coordinates that violate constraints, such as non-alignment, overlap, or out-of-bounds locations. We respectively introduce the multi-modalities, layer decision, and reward function design in Sec. 3.1, Sec. 3.2 and Sec. 3.3.

### 3.1 Multi-Modality Representation of 3D Floorplanning

The state space contains three modalities: FP vision, netlist graph, and block placing sequence.

#### 3.1.1 Vision Modality

It represents current floorplanning through images. We utilize four vision masks to depict chip layout.

**Alignment Mask.** $f_a \in \mathbb{N}^{W \times H}$ ($\mathbb{N}$ is the field of natural number) is a matrix to evaluate the alignment area between blocks $b_i$ and its alignment partner $b_j$. If $b_j$ has already been placed, $f_a[x, y]$ is the intersection area on a 2D projected plane if $b_i$ is placed at $(x, y)$. If $b_j$ has not been placed yet, the

alignment mask of $b_i$ will be set to a matrix full-filled with $\mathrm{aln}_m(i,j)$. The native approach has the complexity $\mathcal{O}(WH)$. However, when $W$ or $H$ is large, it has a low efficiency. We design an efficient alignment mask generation algorithm with *meshgrid* operation, which only iterates in regions causing projection overlap between these blocks and harnessing the power of parallel computing, thus reducing the complexity to $\mathcal{O}(wh)$ namely $\mathcal{O}(1)$. Demonstration of alignment and details of the algorithm are shown in Fig. 2 and Alg. 1. Alignment mask is also utilized to filter out the positions that do not satisfy the alignment constraint.

**Canvas Mask.** Canvas mask $f_c \in \mathbb{N}^{|\mathcal{D}| \times W \times H}$ is a global observation of current chip layout. The canvas mask is initialized to all zeros. After placing a block $b$ at each step, we modify the canvas mask with $f_c[z, x : x+w, y : y+h] \mathrel{+}= 1$. Based on the canvas mask, we can implement a fast calculation of the total overlap of chip layout, shown in Alg. 4 in Appendix G.1.

**Wire Mask & Position Mask [28].** Wire mask $f_w \in \mathbb{N}^{W \times H}$ is a matrix for how HPWL will increase if a block is placed at the position. It records the increase of HPWL by placing the current block to all candidate positions. Position mask $f_p \in \{0, 1\}^{W \times H}$ indicates available positions for current block to place without overlap

---

**Algorithm 1:** Alignment mask generation.

**Input:** Current block $b_i$, alignment partner $b_j$, chip width and height $W, H$
**Output:** Alignment mask $f_a^{(i)}$ for block $b_i$
$x_s = \max(0, x_j - w_i), x_e = \min(x_j + w_j, W)$
$y_s = \max(0, y_j - h_i), \ y_e = \min(y_j + h_j, H)$
$\mathbf{x}_i = arange(x_s, x_e), \ \ \mathbf{y}_i = arange(y_s, y_e)$
$\mathbf{X}_i, \mathbf{Y}_i = meshgrid(\mathbf{x}_i, \mathbf{y}_i)$
$\mathbf{U}_i = \mathbf{X}_i, \ \mathbf{V}_i = \mathbf{X}_i + w_i$
$\mathbf{U}_j = x_j, \ \mathbf{V}_j = x_j + w_j$ // broadcast to a matrix
$\mathbf{U} = where(\mathbf{U}_i > \mathbf{U}_j, \mathbf{U}_i, \mathbf{U}_j)$
$\mathbf{V} = where(\mathbf{V}_i < \mathbf{V}_j, \mathbf{V}_i, \mathbf{V}_j)$
$f_{ax}^{(i)} = \max(0, \mathbf{V} - \mathbf{U})$
// we omit y-dimension due to page limitation
$f_a^{(i)} = zeros(W, H), \ f_a^{(i)}[\mathbf{X}_i, \mathbf{Y}_i] = f_{ax}^{(i)} \odot f_{ay}^{(i)}$

---

or out-of-boundary. 1 implies this position is feasible to place the block.

Overall, in step $t$, given current block $b_t$, we concatenate $f_a^{(t)}, f_c^{(t)}, f_w^{(t)}, f_p^{(t)}$. We also incorporate masks $f_w^{(t+1)}, f_p^{(t+1)}$ of header block in FIFO queue $q$ (introduced in sequence modality in Sec. 3.1.2) in each die, which represent all possible choices for next block in step $t + 1$, providing policy with future horizon. All of these input masks constitute the floorplanning vision modality.

### 3.1.2 Graph and Sequence Modality

**Graph Modality.** Given a netlist, we convert it to a graph $G(V, E)$, where $V$ is the set of vertices and $E$ is the set of edges. For two blocks $b_i, b_j$ within a net, we add edges $e_{ij}, e_{ji}$ between vertex pair $(v_i, v_j)$. For the vertex feature, We select $(bid, x, y, z, w, h, a, p)$, where $bid$ is the block index and $p$ indicates whether this block has already been placed or not. Considering each block has a placing order showing the property as a sequence, it is natural to utilize positional encoding [31] to model this feature. Furthermore, we employ a Graph Attention Network [32] to produce embeddings of graph and nodes, representing the logical connection among blocks.

**Sequence Modality.** Given a die $d_i$, blocks in $d_i$ are sorted by their area in descending order, forming a FIFO (first in, first out) queue $q_i$. Thus, the block placing order for each die is pre-determined, and each queue can be viewed as a sequence. Combined with block features, we enhance and re-organize the multi-die sequence $\mathbf{S} \in \mathbb{R}^{|\mathcal{D}| \times L \times C}$, where $L$ is the maximum number of blocks in a single die, $C$ is the number of features. We select $(bid, x, y, z, w, h, a, p)$ as sequence features. The sequence modality provides the model with global observation of entire block placing order, and is employed in the asynchronous layer decision discussed in Sec. 3.2.

### 3.2 Asynchronous Layer Decision

In 2D FP, blocks are arranged on a single die and are organized in a FIFO queue representing the placing order. However, in 3D scenarios, multiple queues exist due to the multi-die property, necessitating a layer decision mechanism to merge these queues and determine the comprehensive placing order. A native approach is *synchronous* block placing, in which we initially place all blocks on the first die, followed by blocks on the second die, and continue in this manner until the last die. In [26, 27, 28, 30] with synchronous placing, the entire placing order keeps fixed. However, this synchronous die-by-die placing order neglects the cross-die connections and alignment requirements, leading to a relatively poor FP layout. To address this issue, we propose an *asynchronous* layer decision mechanism, which determines the layer for accessing next block. If the policy selects die $d_i$ for next step, the header block of the FIFO queue $q_i$ will be popped as the next block to place.

During the training process under asynchronous block placing, on one hand, the entire block placing order could vary hugely and become unstable due to the sparsity of the reward, making it non-trivial for position decider to determine the positions of blocks. On the other hand, the layer decision module could converge too fast to mode collapse (a fixed placing order leading to poor quality), empirically degenerating to synchronous die-by-die block placing. In our analysis, it is the short forward horizon leading to this problem. Since policy is only able to sense current and next states, it lacks the global receptive field of entire placing order.

To address these issues, we enhance the representation with the sequence modality, and employ Transformer [31] to extract global placing order feature. Given the multi-die sequence feature $\mathbf{S} \in \mathbb{R}^{|\mathcal{D}| \times L \times C}$, self-attention is computed. $\mathbf{S}$ is viewed as the source sequence input with length $L$. The memory $\mathbf{M} \in \mathbb{R}^{|\mathcal{D}| \times L \times L}$, output of self-attention of $\mathbf{S}$ is given by [31]:

$$\mathbf{M} = \mathrm{LN}(\mathrm{MSA}(\mathbf{S}) + \mathbf{S}), \mathbf{M} = \mathrm{LN}(\mathrm{FFN}(\mathbf{M}) + \mathbf{M}), \tag{3}$$

where MSA is the multi-head self-attention, LN is layer normalization, and FFN is the feed-forward network. Next, cross-attention is applied on $\mathbf{M}$ and current block feature $\mathbf{b}_k$. The single block feature $\mathbf{b}_k \in \mathbb{R}^{1 \times C}$ is treated as the target sequence with length 1, serving as the query vector. The output of cross-attention $\mathbf{O} \in \mathbb{R}^{|\mathcal{D}| \times 1 \times C}$ is [31]:

$$\mathbf{O} = \mathrm{LN}(\mathrm{MHA}(\mathbf{b}_k, \mathbf{M}, \mathbf{M}) + \mathbf{b}_k), \mathbf{O} = \mathrm{LN}(\mathrm{FFN}(\mathbf{O}) + \mathbf{O}), \tag{4}$$

where $\mathrm{MHA}(\mathbf{Q}, \mathbf{K}, \mathbf{V})$ denotes multi-head attention. Finally, the next layer decision probability vector can be further computed through linear projection and $\mathrm{softmax}$ operation.

## 3.3 Reward Function with Local Advantage and Global Baseline

In 3D floorplanning task, final wirelength, overlap and alignment can only be accessed at the end of each episode, leading to a sparse reward. GraphPlace [26] uses this sparse reward design, where rewards at intermediate steps are all zeros. DeepPlace [27] adopts it with additional intrinsic reward via Random Network Distillation [33]. MaskPlace [28] introduces a dense reward scheme based on partial HPWL, which is calculated only on the currently placed blocks at each step. However, in two former methods [26, 27], they fail to accurately sense the intermediate quality through reward, and in MaskPlace [28], only difference between local

---

**Algorithm 2:** Reward function with local advantage and global baseline.

**Input:** (Partial) Alignment score aln, (partial) overlap $o$, (partial) HPWL, and corresponding weight $w_a, w_o, w_l$
**Output:** Reward $r$ for each step
**for** $t$ **from** $\mathrm{len}(episode)$ **to** 1 **do**
  **if** $t$ *is the end of an episode* **then**
    $r_t = w_a \cdot \mathrm{aln}_t - w_o \cdot o_t - w_l \cdot \mathrm{HPWL}_t$
    $b = r_t$
  **else**
    $r_t = w_a \cdot (\mathrm{aln}_t - \mathrm{aln}_{t-1}) - w_o \cdot (o_t - o_{t-1}) - w_l \cdot (\mathrm{HPWL}_t - \mathrm{HPWL}_{t-1}) + b$

---

adjacent steps is involved, lacking the global view of the whole episode. As a result, all these reward designs demonstrate relatively poor performances, especially with the complicated hybrid action space consisting of position, layer and aspect ratio.

To alleviate this problem, we design a novel reward function with local advantage and global baseline. We define *local advantage* as the difference of metric between two adjacent steps, and *global baseline* as the overall metric at the end of an episode. With local advantage, our model has the ability to acquire current state is whether better or worse than the previous. Global baseline depicts the overall quality of the final floorplanning result. It is also essential for asynchronous layer decision module to avoid early convergence and degeneration to poor die-by-die synchronous block placing order, shown in Sec. 4.4 and Fig. 6b. Details of reward design is shown in Alg. 2.

## 3.4 Flexible RL with Hybrid Action Space for 3D Floorplanning

We employ RL with a hybrid action space to address the 3D FP task. The state space consists of three modalities, including floorplanning vision, netlist graph and sequence of block placing order. The hybrid action space is formulated as $\mathcal{X} \times \mathcal{Y} \times \mathcal{Z} \times \mathcal{R}$, with $\mathcal{X}, \mathcal{Y}, \mathcal{Z}$ as discrete sets and $\mathcal{R}$ as a continuous set. In each step $t$, the policy outputs three distributions: **1)** a discrete probability distribution for 2D position $(x_t, y_t)$ of current block $b_t$, **2)** a discrete probability distribution to determine the layer $z_{t+1}$ for accessing next block $b_{t+1}$, and finally **3)** the mean value and standard deviation of a contiguous Gaussian distribution to determine the aspect ratio $\mathrm{AR}_{t+1}$ for $b_{t+1}$. The

Table 2: Alignment score comparison among baselines and our method. The higher the alignment score, the better, and the optimal results are shown in **bold**. C/M means Circuit/Method.

| C/M | 3D-B*-SA [21] | RL-CBL [24] | Wiremask-BBO [30] | GraphPlace [26] | DeepPlace [27] | MaskPlace [28] | Ours |
|---|---|---|---|---|---|---|---|
| ami33 | 0.550±0.058 | 0.132±0.038 | 0.179±0.091 | 0.207±0.067 | 0.286±0.051 | 0.300±0.017 | **0.905±0.017** |
| ami49 | 0.438±0.099 | 0.107±0.043 | 0.222±0.082 | 0.265±0.063 | 0.180±0.056 | 0.218±0.052 | **0.955±0.010** |
| n10 | 0.383±0.167 | 0.241±0.076 | 0.211±0.004 | 0.197±0.049 | 0.235±0.080 | 0.354±0.066 | **0.917±0.012** |
| n30 | 0.537±0.159 | 0.108±0.040 | 0.288±0.051 | 0.233±0.039 | 0.287±0.074 | 0.511±0.067 | **0.920±0.024** |
| n50 | 0.626±0.158 | 0.048±0.016 | 0.290±0.053 | 0.378±0.120 | 0.343±0.064 | 0.764±0.002 | **0.970±0.004** |
| n100 | 0.131±0.051 | 0.016±0.008 | 0.195±0.034 | 0.279±0.050 | 0.332±0.073 | 0.575±0.046 | **0.961±0.017** |
| n200 | 0.033±0.025 | 0.013±0.009 | 0.182±0.031 | 0.360±0.060 | 0.387±0.039 | 0.534±0.041 | **0.923±0.020** |
| n300 | 0.009±0.009 | 0.005±0.005 | 0.205±0.038 | 0.383±0.023 | 0.399±0.031 | 0.533±0.023 | **0.965±0.010** |
| Avg. | 0.338 | 0.084 | 0.222 | 0.288 | 0.306 | 0.474 | **0.940** |

Table 3: HPWL (the lower the better) comparison. The optimal results are shown in **bold**.

| C/M | 3D-B*-SA [21] | RL-CBL [24] | Wiremask-BBO [30] | GraphPlace [26] | DeepPlace [27] | MaskPlace [28] | Ours |
|---|---|---|---|---|---|---|---|
| ami33 | 85,162±5,563 | 85,303±4,147 | 66,387±3,531 | 82,685±6,271 | 79,457±6,885 | 62,125±829 | **58,339±1,894** |
| ami49 | 1,400,787±68,043 | 1,338,219±98,556 | 1,100,891±95,467 | 1,455,872±80,468 | 1,356,203±56,434 | 1,128,110±90,645 | **762,712±12,878** |
| n10 | 35,230±115 | 34,805±1,132 | 33,046±36 | 36,520±869 | 34,371±817 | 33,648±1,096 | **29,781±75** |
| n30 | 101,672±2,665 | 105,796±1,844 | 87,198±1,862 | 98,437±1,860 | 97,293±3,192 | 86,291±764 | **83,962±1,030** |
| n50 | 132,421±3,123 | 156,113±4,079 | 111,878±3,371 | 136,980±2,219 | 126,910±2,836 | 113,145±407 | **105,839±916** |
| n100 | 223,381±9,123 | 275,982±10,348 | 181,572±1,966 | 209,940±4,161 | 223,359±5,330 | 189,100±2,133 | **176,375±960** |
| n200 | 422,060±9,101 | 572,649±37,831 | 325,453±3,488 | 402,650±6,117 | 418,348±5,134 | 375,250±2,939 | **316,199±1,080** |
| n300 | 633,344±5,513 | 990,465±37,719 | 467,906±5,362 | 596,615±4,353 | 635,165±9,636 | 532,087±4,214 | **459,221±5,935** |
| Avg. | 379,257 | 444,917 | 296,791 | 377,462 | 371,388 | 314,969 | **249,053** |

reason for action $a_t$ containing the layer and aspect ratio for next step $t+1$ instead of $t$ is that: after the execution of $a_t$, next state $s_{t+1}$ can be generated only after block $b_{t+1}$ to place at step $t+1$ and its shape have already been determined, since $b_{t+1}$ and its shape $(w_{t+1}, h_{t+1})$ are involved in the calculation of alignment mask and wire mask. To guarantee adherence to the specified non-overlap and alignment constraints, position mask $f_p$ and alignment mask $f_a$ are incorporated in 2D position decision. Only $(x, y)$ satisfying $f_p[x, y] = 1$ and $f_a[x, y] \geq \mathrm{aln}_m$ are considered as valid positions, where $\mathrm{aln}_m$ is the required minimum alignment area between current block and its alignment partner. Finally, without the reliance on conventional heuristic FP representation, our approach exhibits more flexibility to directly solve position, aspect ratio and cross-die alignment for blocks.

We select the Actor-Critic [34] framework and Hybrid Proximal Policy Optimization [35, 36] algorithm. The objective function of our hybrid policy $\pi_\theta(a_t|s_t)$ can be formulated as:

$$L(\theta) = \sum_{k=1}^{3} \lambda_k \cdot \hat{\mathbb{E}}_t \left[ \min \left( r_t^{(k)}(\theta)\hat{A}_t, \mathrm{clip}\left( r_t^{(k)}(\theta), 1 - \varepsilon, 1 + \varepsilon \right) \hat{A}_t \right) \right], \quad (5)$$

where $k = 1, 2, 3$ represents position, layer and aspect ratio decision. $\lambda_k$ is the weight for each clip loss. $r_t^{(k)}(\theta)$ is the probability ratio $\frac{\pi_\theta(a_t^{(k)}|s_t)}{\pi_{\theta_{\mathrm{old}}}(a_t^{(k)}|s_t)}$. $\hat{A}_t$ denotes the generalized advantage estimation (GAE) [37], and $G_t = \hat{A}_t + V_t$ is the cumulative discounted reward [37, 38]. $V_t$ is the estimated state value from critic network $V_\phi(s_t)$, and critic network is updated with minimizing Mean Squared Error (MSE) $L(\phi) = \lambda_\phi \cdot \hat{\mathbb{E}}_t \left[ (G_t - V_\phi(s_t))^2 \right]$. Entropy of each action distribution is also added as a regularization term for exploration encouragement. Training algorithm is shown in Appendix G.2, and details of model architecture are shown in Appendix D.

## 4 Experiment and Analysis

### 4.1 Evaluation Protocol and Benchmark

We evaluate the performance of FlexPlanner and other typical methods on public benchmark **MCNC** [3] and **GSRC** [4] shown in Table 7 in Appendix B. The floorplanning region is set to square region. I/O ports are projected to the fixed outline, and their positions are kept unchanged. Aspect ratio of each block can vary in range $\left[\frac{1}{2}, 2\right]$. Each experiment is run for five times with different seeds. More implementation details and hyper-parameter settings can be found in Appendix C.

---

[3] http://vlsicad.eecs.umich.edu/BK/MCNCbench/
[4] http://vlsicad.eecs.umich.edu/BK/GSRCbench/

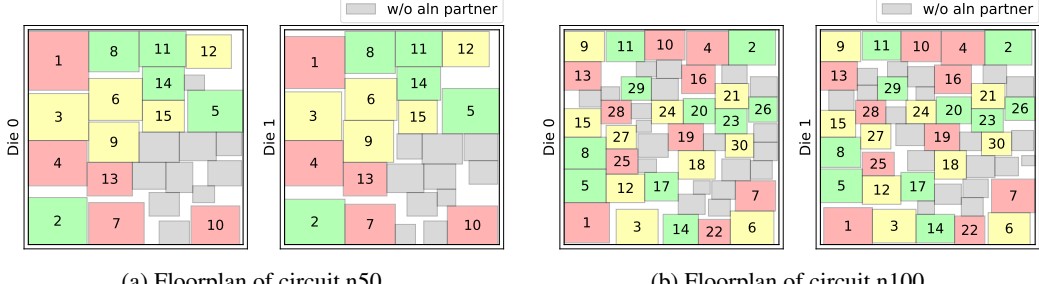

(a) Floorplan of circuit n50              (b) Floorplan of circuit n100

Figure 3: Our 3D floorplan result. Two blocks with the same index and the same color on different dies form an alignment pair, which roughly locate on the same positions and share a 2D common projected area. Gray blocks mean they do not have alignment partners.

## 4.2 Comparison with Baselines

Methods including heuristic-based and learning-based approaches are selected as baselines, and the implementation details are shown in Appendix E. Average alignment score and total HPWL are employed as evaluation metrics. Results are shown in Table 2 and 3. For alignment score, our approach achieves 0.940, significantly surpassing the second-best method which scores 0.474. For HPWL, our approach achieves an average reduction of 16%. Our method reduces HPWL to 249,053, and the second-best method (Wiremask-BBO) reduces it to 296,791. However, it only achieves 0.222 for alignment score, failing to effectively tackle the alignment constraint, empirically showing that our approach is capable of address alignment and HPWL simultaneously. Demonstration of 3D floorplanning results by our method are also shown in Fig. 3 and Fig. 4. We also compare performances on overlap, out-of-bound and runtime, shown in Appendix F. Capability and flexibility of our approach to address pre-placed modules (PPMs) are also demonstrated in Appendix F.5.

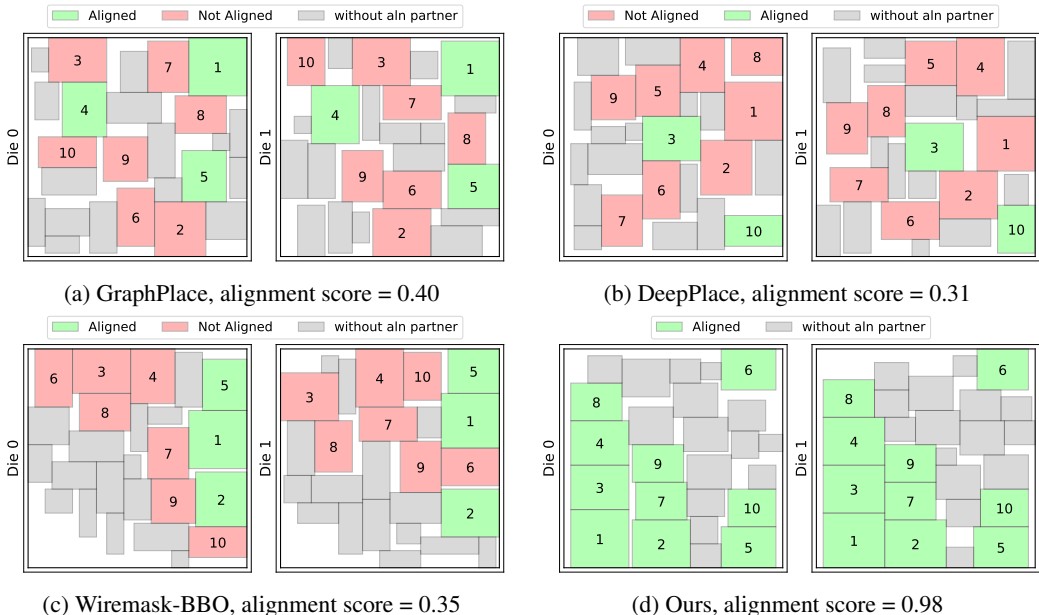

(a) GraphPlace, alignment score = 0.40       (b) DeepPlace, alignment score = 0.31

(c) Wiremask-BBO, alignment score = 0.35       (d) Ours, alignment score = 0.98

Figure 4: Visualization of cross-die block alignment on circuit n50. Two blocks with the same index forms an alignment pair. For a pair with block $i, j$, we calculate individual alignment score $\mathrm{aln}(i, j)$ according to Eq. 1. Green means these two blocks are aligned ($\mathrm{aln}(i, j) \geq 0.5$) while red means not aligned ($\mathrm{aln}(i, j) < 0.5$). Total alignment score is calculated according to Alg. 3 in Appendix G.1. It demonstrates that our method achieves much better alignment score than other baselines.

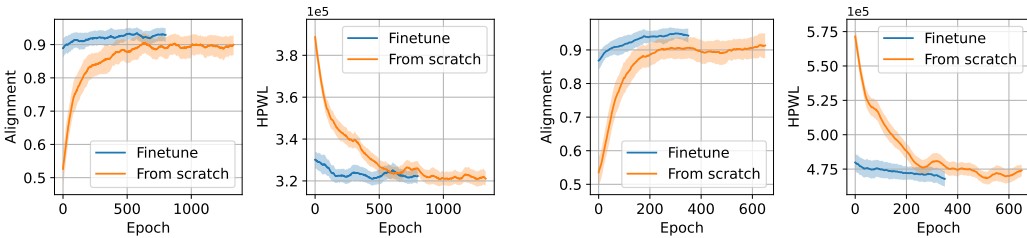

(a) Training curve on circuit n200.  (b) Training curve on circuit n300.

Figure 5: Training curve between fine-tune (based on circuit n100) and training from scratch.

Table 4: Zero-shot transferability evaluation (training on circuit n100).

| Metric/Circuit | | ami33 | ami49 | n10 | n30 | n50 | n200 | n300 |
|---|---|---|---|---|---|---|---|---|
| Alignment (↑) | value | 0.859 | 0.894 | 0.875 | 0.947 | 0.970 | 0.877 | 0.908 |
| | ratio | 0.949 | 0.936 | 0.954 | 1.029 | 1.000 | 0.950 | 0.941 |
| HPWL (↓) | value | 59,923 | 835,170 | 30,720 | 87,784 | 111,039 | 322,242 | 462,780 |
| | ratio | 1.027 | 1.095 | 1.032 | 1.046 | 1.049 | 1.019 | 1.008 |

Table 5: Ablation study on n100. sync: synchronous die-by-die placing. w/o aln: remove alignment mask in vision modality. w/o seq: remove sequence modality. w/o graph: remove graph modality. sparse rew: the same reward as GraphPlace [26]. diff rew: the same reward as MaskPlace [28].

| Metric/Method | sync | w/o aln | w/o graph | w/o seq | sparse rew | diff rew | Ours |
|---|---|---|---|---|---|---|---|
| Alignment | 0.850±0.042 | 0.349±0.026 | 0.874±0.027 | 0.860±0.021 | 0.840±0.018 | 0.721±0.054 | **0.961±0.017** |
| HPWL | 185,624±1,822 | 187,617±3,096 | 185,079±1,681 | 186,005±846 | 189,219±1,164 | 190,384±2,192 | **176,639±1,001** |

## 4.3 Transferable 3D Floorplanning

The zero-shot transferability of our method is also evaluated. We firstly train the model on circuit n100, and directly test the performance without any fine-tuning. In Table 4, ratio is calculated between running inference and training on corresponding circuit. It demonstrates that our model exhibits a good zero-shot transferability, on either smaller or larger cases. The transferability is also demonstrated through the training curves between fine-tuning (based on pre-trained weights of circuit n100) and training from scratch. In Fig. 5, through fine-tune technique, better or similar performance can be achieved, conserving substantial training resources.

## 4.4 Ablation Studies

We ablate effectiveness of each component in our approach, including asynchronous layer decision, multi-modality representation and reward function. Experiments on circuit n100 are shown in Table 5. Removing the alignment mask leads to a huge drop of alignment score, from 0.961 to 0.349. We also evaluate the effectiveness of incorporating the alignment mask as an input feature and a constraint. In Fig. 6a, 'input' refers to the scenario where we feed the alignment mask as input to the model, while 'constr.' indicates that the alignment mask is utilized to filter out invalid positions that do not satisfy the alignment constraint. As the input, alignment mask plays a critical role to effectively capture the alignment information. As the constraint, it reduces the action space and accelerates training process.

Shown in the column 'sync' in Table 5, fixed synchronous die-by-die block placing order performs worse than asynchronous layer decision, since in the latter, more flexibility is provided to plan the entire placing order. Our reward design, incorporating the local advantage and global baseline, is crucial for fully leveraging the asynchronous layer decision mechanism. In Fig. 6b, with the guidance of our reward, the layer decision module keeps active to learn an optimal placing order. However, with other reward design schemes, it is faced with either rapid convergence to the degeneration of synchronous die-by-die decision-making, or unstable oscillation. Under reward function 'diff' in MaskPlace [28], only the difference between two consecutive steps is focused, while the global reward is disregarded, leading to rapid convergence to the degeneration of synchronous die-by-die layer decision. Under reward function 'sparse' in GraphPlace [26], rewards for intermediate steps are all zeros, and only the reward of the final step is non-zero. The policy lacks the modeling of local information, resulting in its inability to accurately assess the influence of current action on the entire decision-making process.

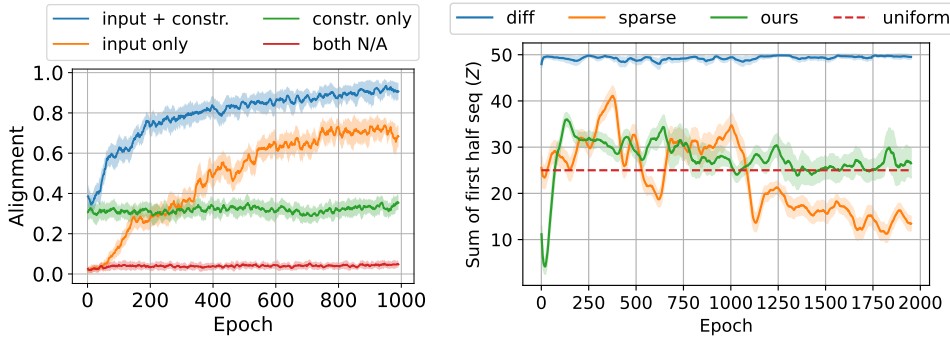

(a) Effectiveness of the alignment mask.   (b) Effectiveness of rewards on layer decision.

Figure 6: (a) Effectiveness of the alignment mask. As the input feature, it plays a critical role in capturing the alignment information. As the constraint (constr.), it reduces the action space and accelerates the training process. (b) Effectiveness of rewards on layer decision, shown in circuit n100 with episode length $L = 100$ and $|\mathcal{D}| = 2$ dies. $z_t$ is the determined layer index at step $t$, and we note $Z = \sum_{t=1}^{L/2} z_t$. $Z \to 0$ or $Z \to L/2$ means degeneration to die-by-die synchronous layer decision (almost all die 0 or 1 in the first half episode).

## 5 Related Works

Classical methods in FP can be roughly categorized into heuristics and analytical approaches. The former typically models FP with a certain representation, such as B*-tree [17], Corner Block List [18] and Sequence Pair [39], based on which heuristic algorithm (especially Simulated Annealing [40]) is utilized to search for an optimal solution. Finally, the representation is converted to corresponding FP result via a decoding scheme. Apart from the heuristics, analytical approaches [9, 22, 41] regard the FP problem as an electrostatic system [42, 43]. They compute the gradient of objective functions w.r.t. block coordinates, and utilize gradient descent-based algorithm to optimize the solution.

**RL-based floorplanning approaches.** Recently, floorplanning also attracted attention from the reinforcement learning communities. For instance, [23, 24, 25] incorporate traditional FP representation and RL. Among these methods, at each step, the policy network either 1) decides whether to accept the randomly perturbed state or not [23, 25] or 2) determines how to perturb the current state [24]. [26, 27, 28] primarily focus on making decisions about block position in 2D scenarios with RL. GraphPlace [26] and DeepPlace [27] incorporate graph and vision as representation. MaskPlace [28] employs visual representation, and designs a dense reward function based on partial HPWL.

**RL with hybrid action space.** Common RL can be categorized into discrete and continuous action spaces. However, in certain scenarios, besides the discrete action, we need to make decisions regarding its parameters, typically residing in continuous spaces [36]. Consequently, it gives rise to a hybrid action space. [44] proposes to discretize the continuous portion, which creates a large discrete set and sacrifices fine-grained control. Alternatively, [45, 46, 47, 48] convert the discrete action selection into a continuous space with an actor network and employ a DQN-based [49, 50] algorithm for training, but found to be unstable and inefficient. [36] suggests to use multiple policy heads consisting of one for discrete actions and the others for corresponding continuous parameters separately. [51] proposes to encode the hybrid action space to a continuous space with GAE [37].

## 6 Conclusion

In this paper, we propose FlexPlanner, a flexible learning-based method in hybrid action space with multi-modality representation for 3D floorplanning task. It involves no reliance on heuristic-based search, thus achieves better flexibility to tackle position, aspect ratio and cross-die alignment for blocks under complex constraints. FlexPlanner outperforms baselines in alignment score and wirelength, and it also demonstrates zero-shot transferability on unseen circuits. This paper also has some *limitations* for future work: further optimization in 3D FP could be involved such as thermal optimization. Our method has no potential harm to the public society at the moment.

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

# A  Notation

All notations in this paper are shown in Table 6.

Table 6: Notation.

| Notation | Meaning |
|---|---|
| $b$ | block |
| $t$ | I/O port (terminal) |
| $d$ | die/layer |
| $\mathcal{D}$ | the set of all dies/layers |
| $w, h$ | width, height of a block |
| $a$ | area of a block |
| AR | aspect ratio of a block |
| $x, y$ | 2D coordinate of a block/port |
| $z$ | the layer/die index of a block/port |
| $W, H$ | width, height of die/layer |
| $o_{ij}$ | overlap area between block $b_i, b_j$ |
| $\mathrm{aln}(i, j)$ | alignment score between block $b_i, b_j$ |
| $\mathrm{aln}_m(i, j)$ | required minimum alignment area between block $b_i, b_j$ |
| $q_i$ | the block placing order FIFO queue of die $d_i$ |
| $f_a$ | alignment mask |
| $f_c$ | canvas mask |
| $f_w$ | wire mask |
| $f_p$ | position mask |

# B  Statistics of Benchmark

We evaluate the performance of FlexPlanner and other typical methods on public benchmark MCNC [5] and GSRC [6] shown in Table 7. The 'alignment' means the number of blocks with alignment partner.

Table 7: MCNC and GSRC benchmark

| circuit | block | I/O port | net | alignment |
|---|---|---|---|---|
| ami33 | 33 | 40 | 121 | 20 |
| ami49 | 49 | 22 | 396 | 20 |
| n10 | 10 | 69 | 118 | 10 |
| n30 | 30 | 212 | 349 | 20 |
| n50 | 50 | 209 | 485 | 30 |
| n100 | 100 | 334 | 885 | 60 |
| n200 | 200 | 564 | 1585 | 60 |
| n300 | 300 | 569 | 1893 | 60 |

# C  Implementation Details

## C.1  Computational Resources

We use PyTorch [52] deep learning framework and tianshou [38] Reinforcement Learning framework. We select Adam [53] optimizer with learning rate 0.0001. We train and test our model on a Linux server with one NVIDIA GeForce RTX 3090 GPU with 24 GB CUDA memory, two AMD Ryzen Threadripper 3970X 32-Core Processors at 3.70 GHz and 128 GB RAM.

---

[5] http://vlsicad.eecs.umich.edu/BK/MCNCbench/
[6] http://vlsicad.eecs.umich.edu/BK/GSRCbench/

## C.2 Minimum Alignment Requirement Configuration

For the alignment setting, given an alignment pair $(b_i, b_j)$, the minimum alignment requirement $\text{aln}_m(i,j)$ is set to $\text{aln}_m(i,j) = \alpha_{ij} \cdot \min\{a_i, a_j\}$, where $a_i, a_j$ are the area of block $b_i, b_j$. The coefficient $\alpha_{ij}$ controls the minimum alignment requirement between block $b_i, b_j$, and can be adjusted according to the specific circumstances for each alignment pair. In our experiments, we set *all* $\alpha_{ij} = 1.0$, which is the most challenging scenario to evaluate the effectiveness of our approach.

## C.3 Hyper-Parameters

Other hyper-parameters are shown in Table 8.

Table 8: Hyper-parameters.

| Argument | Value |
| --- | --- |
| learning rate | 0.0001 |
| parallel environments $n_e$ | 8 |
| buffer size | $n_e \times \text{len}(episode)$ |
| reward weight for alignment $w_a$ | 0.5 |
| reward weight for HPWL $w_l$ | 1.0 |
| reward weight for overlap $w_o$ | 0.5 |
| area utilization | 85% |
| clip loss weight for position decision $\lambda_1$ | 1.0 |
| clip loss weight for layer decision $\lambda_2$ | 1.0 |
| clip loss weight for ratio decision $\lambda_3$ | 0.5 |
| range of block aspect ratio | $\left[\frac{1}{2}, 2\right]$ |
| batch size | 128 |
| die width $W$ | 128 |
| die height $H$ | 128 |
| number of dies $|\mathcal{D}|$ | 2 |
| value loss weight $\lambda_\phi$ | 0.5 |
| number of epochs | 1,000 |
| number of update epochs | 10 |
| clip $\varepsilon$ | 0.2 |
| reward discount factor $\gamma$ | 0.99 |
| GAE $\lambda$ | 0.95 |

# D Model Architecture

We introduce the networks in our pipeline shown in Fig. 1 as follows:

**Shared Encoder.** The shared encoder $E_\phi$ mainly consists of two parts: 1) a CNN-based backbone for floorplanning vision modality input, and 2) a GNN (Graph Neural Network)-based backbone Graph Attention Network [32] with two layers for graph modality.

In step $t$, given current block $b_t$, we concatenate the alignment mask $f_a^{(t)}$, canvas mask $f_c^{(t)}$, wire mask $f_w^{(t)}$, and position mask $f_p^{(t)}$ together. We also incorporate $f_w^{(t+1)}, f_p^{(t+1)}$ of header block in FIFO queue $q$ of each die. These masks of queue header blocks represent all possible choices in step $t+1$, providing policy with future horizon. All of these input masks constitute the floorplanning vision modality.

Taking the netlist graph as input, the GNN backbone outputs nodes embeddings, representing the local receptive field information. Besides, a global average pooling layer is applied on these node embeddings to calculate the graph embedding, which captures the global view of the whole netlist graph.

**Critic Network.** The critic network $V_\phi$ mainly consists of three parts: 1) the shared encoder $E_\phi$, 2) a sequence Transformer [31] with two encoder layers and two decoder layers for block placing order sequence, and 3) a die/layer embedding model to represent each die/layer with a learnable feature vector. The shared encoder $E_\phi$ is responsible for processing the floorplanning vision input, and the Transformer is for the input sequence modality. Serving as the source sequence in Transformer, the multi-die sequence feature $\mathbf{S} \in \mathbb{R}^{|\mathcal{D}| \times L \times C}$ is employed in the calculation of memory, which is the output of self-attention. And the single block feature

$\mathbf{b}_k \in \mathbb{R}^{1 \times C}$ is treated as the target sequence with length 1, serving as the query in cross-attention. To Stabilize training, the shared encoder $E_\phi$ is only updated during the updating process of critic network. As a result, the critic network is responsible to take input as the state $s_t$ and output corresponding state value $v_t$.

**Policy Network.**

- **Position Decider.** The position decider network $\pi_\theta^{(1)}$ mainly consists of a generator network. It takes input as the output of the shared encoder $E_\phi$, and outputs a 2D feature map as the probability matrix to determine the 2D position for current block. We borrow the generator architecture from InfoGAN [54], which utilizes up-sampling layer instead of transposed convolutional layer to realize scaling-up. It contains three blocks and each block is with convolutional layer, batch normalization, leaky ReLU [55] activation and up-sampling layer. Finally, it outputs a probability matrix $\mathbf{M}_p^{(t)} \in \mathbb{R}^{W \times H}$ to determine the position $(x_t, y_t)$ for current block $b_t$.

- **Layer Decider.** The layer decider network $\pi_\theta^{(2)}$ mainly consists of three parts: 1) a CNN-based model for FP vision modality, 2) a die embedding model to represent current input die, and 3) a Transformer network to process the input block placing order sequence for each die. The CNN model consists of three blocks, and each block has one convolutional layer, one max pooling layer, with ReLU as activation function. The Transformer model consists of two encoder layers and two decoder layers, and is designed to process the input multi-die block placing order sequence. The layer decider network takes input as the state $s_t$ and outputs a probability vector $\mathbf{p}_z^{(t)} \in \mathbb{R}^{|\mathcal{D}|}$ to determine the layer $z_{t+1}$ for accessing next block $b_{t+1}$.

- **Aspect Ratio Decider.** The aspect ratio decider network $\pi_\theta^{(3)}$ mainly has a CNN model, consisting of three 2D convolutional layers with ReLU activation function and max pooling layer. It takes input as the floorplanning vision modality in state $s_t$ and the feature of next block $b_{t+1}$ to determine the aspect ratio $\text{AR}_{t+1}$ for $b_{t+1}$. Since the aspect ratio is an action in contiguous space, we select Gaussian distribution to depict it, and the network outputs corresponding mean value and standard deviation.

# E  Baselines

The baselines referred in Sec. 4 are introduced as follows:

**3D-B\*-SA** [21] is a heuristic-based approach, which represents a 3D floorplanning by B\*-tree [17]. Simulated Annealing (SA) [40] is selected to search an optimal result. At each iteration, current B\*-tree is randomly perturbed to a new state, which will be accepted based on a certain probability. Finally, the B\*-tree is converted to corresponding floorplannning result via a decoding scheme. In previous scenarios, only HPWL and out-of-bound penalty are incorporated into heuristics (cost/energy function). To optimize the alignment in 3D FP, we also combine the alignment score into its heuristics to guide the solution searching.

**RL-CBL** [24] combines heuristic-based search Corner Block List (CBL) [18, 19] and Reinforcement Learning together, utilizing policy network to determine how to perturb current FP at each step. It utilizes intermediate out-of-bound penalty and final HPWL as reward function. In 3D FP with multiple stacked layers, we extend CBL to 3D scenario. In order to realize the optimization of cross-die block alignment, we also incorporate alignment score into reward function. However, it is incapable of addressing the variable aspect ratio of soft blocks.

**Wiremask-BBO** [30] is a black-box optimization (BBO) framework, by using a wire-mask-guided [28] greedy procedure for objective evaluation. At each step, it utilize the position mask to filter out invalid positions causing overlap or out-of-bound. Within the valid positions, the location with the minimum increment of HPWL is selected to place current block. Considering the alignment requirement in 3D FP, we incorporate the alignment mask to further filter out invalid positions which do not satisfy the alignment constraint. However, it is incapable of addressing the variable aspect ratio of soft blocks.

**GraphPlace** [26], **DeepPlace** [27], **MaskPlace** [28] are three learning-based approaches, utilizing Reinforcement Learning to decide the position for each block on 2D chip layout. Variable aspect ratios of soft blocks are not taken into consideration. In this task, we extend them to 3D scenarios, and incorporate the alignment mask to filter out invalid positions where alignment constraint is not satisfied. Besides, alignment score is also involved into the calculation of reward function, with the purpose of assisting them to realize the optimization of cross-die alignment.

# F  Addtional Experiments

## F.1  Runtime Comparison

Comparison of runtime among our approach and other baselines are shown in Table 9. The runtime of our method is either lower or comparable to other learn-based methods [26, 27, 28], and is better than the heuristic-based method [21], where tens of thousands of iterations are required. Pipeline of Wiremask-BBO [30] is the same as

MaskPlace [28], while neural network is not involved in it, contributing to the shortest inference time. For the approach RL-CBL [24], it is faster than other learning-based methods since its environment is relatively simple, only responsible for decoding the heuristic representation Corner Block List (CBL) [18, 19] into corresponding FP result. However, the performance is also limited by CBL, and the final FP result is further worse than other methods without the reliance of heuristic representation.

Table 9: Runtime (second) comparison among baselines and our method. The lower the runtime, the better. The unit of runtime is in seconds. C/M means Circuit/Method.

| C/M | 3D-B*-SA [21] | RL-CBL [24] | Wiremask-BBO [30] | GraphPlace [26] | DeepPlace [27] | MaskPlace [28] | Ours |
|---|---|---|---|---|---|---|---|
| ami33 | 18.374 | 1.131 | 0.228 | 3.155 | 1.819 | 4.679 | 2.419 |
| ami49 | 31.917 | 1.308 | 0.426 | 2.782 | 2.736 | 3.593 | 2.219 |
| n10 | 6.679 | 0.965 | 0.094 | 1.813 | 1.330 | 2.682 | 2.026 |
| n30 | 21.944 | 1.122 | 0.262 | 1.867 | 2.456 | 2.256 | 2.709 |
| n50 | 35.385 | 1.323 | 0.427 | 4.778 | 3.599 | 2.342 | 2.533 |
| n100 | 65.971 | 2.575 | 1.027 | 4.408 | 4.259 | 4.593 | 3.975 |
| n200 | 127.140 | 2.583 | 2.477 | 5.915 | 6.187 | 5.475 | 7.991 |
| n300 | 176.222 | 3.599 | 4.492 | 10.225 | 9.521 | 10.128 | 10.261 |
| Avg. | 60.454 | 1.826 | 1.179 | 4.368 | 3.989 | 4.469 | 4.266 |

## F.2 Out-of-Bound

For heuristics-based methods, such as 3D-B*-SA [21] and RL-CBL [24], the heuristic representation will be converted to corresponding floorplan result through a decoding scheme. Although non-overlap can be guaranteed, blocks in the final FP could be out of the fixed outline, leading to the low area utilization of chip die and out-of-bound. In our method FlexPlanner, each block can be naturally placed within the boundary due to the alignment mask and position mask. Consequently, the occurrence of out-of-bound floorplan is effectively prevented. We compare the out-of-bound in Table 10, and the outbound is calculated as follows:

$$x_m = \max_{b_i \in \mathcal{B}} \{x_i + w_i\}$$
$$y_m = \max_{b_i \in \mathcal{B}} \{y_i + h_i\}$$
$$\text{outbound} = \frac{\max\{0, x_m - W\}}{2W} + \frac{\max\{0, y_m - H\}}{2H}. \tag{6}$$

Table 10: Comparison of out-of-bound, the lower the better.

| Circuit/Outbound | 3D-B*-SA [21] | RL-CBL [24] | Ours |
|---|---|---|---|
| ami33 | 0.000±0.000 | 0.087±0.043 | 0.000±0.000 |
| ami49 | 0.000±0.000 | 0.213±0.059 | 0.000±0.000 |
| n10 | 0.006±0.011 | 0.026±0.031 | 0.000±0.000 |
| n30 | 0.000±0.000 | 0.116±0.025 | 0.000±0.000 |
| n50 | 0.000±0.000 | 0.156±0.052 | 0.000±0.000 |
| n100 | 0.000±0.000 | 0.223±0.081 | 0.000±0.000 |
| n200 | 0.008±0.014 | 0.359±0.117 | 0.000±0.000 |
| n300 | 0.014±0.009 | 0.549±0.097 | 0.000±0.000 |
| Average | 0.004 | 0.216 | 0.000 |

## F.3 Overlap

We also compare performance on overlap among our method and baselines, shown in Table 11. Our method reduces the overlap to 0.001, which comparable with 3D-B*-SA [21], RL-CBL [24], and better than other methods. In 3D-B*-SA [21] and RL-CBL [24], although non-overlap can be guaranteed, blocks in corresponding FP could be out of the fixed outline, leading to the low area utilization of chip die and out-of-bound as discussed in Appendix F.2.

## F.4 Influence of Grid Size

Due to the different size of circuits, we discretize the sizes into uniform region. Given a chip die with original shape $W_o \times H_o$, the region is projected into $W \times H$. For each block $b$ with shape $w \times h$, its shape is also projected into $\max\left\{1, \text{round}(w \cdot \frac{W}{W_o})\right\} \times \max\left\{1, \text{round}(h \cdot \frac{H}{H_o})\right\}$. We also empirically investigate the

Table 11: Overlap comparison among baselines and our method. The lower the overlap, the better.

| C/M | 3D-B*-SA [21] | RL-CBL [24] | Wiremask-BBO [30] | GraphPlace [26] | DeepPlace [27] | MaskPlace [28] | Ours |
|---|---|---|---|---|---|---|---|
| ami33 | 0.000±0.000 | 0.000±0.000 | 0.050±0.027 | 0.037±0.025 | 0.024±0.016 | 0.012±0.009 | 0.000±0.000 |
| ami49 | 0.000±0.000 | 0.000±0.000 | 0.000±0.001 | 0.002±0.002 | 0.001±0.001 | 0.009±0.008 | 0.000±0.000 |
| n10 | 0.000±0.000 | 0.000±0.000 | 0.208±0.005 | 0.229±0.047 | 0.112±0.028 | 0.085±0.065 | 0.009±0.008 |
| n30 | 0.000±0.000 | 0.000±0.000 | 0.007±0.007 | 0.042±0.028 | 0.037±0.022 | 0.016±0.005 | 0.000±0.000 |
| n50 | 0.000±0.000 | 0.000±0.000 | 0.000±0.000 | 0.009±0.014 | 0.029±0.020 | 0.004±0.007 | 0.000±0.000 |
| n100 | 0.000±0.000 | 0.000±0.000 | 0.000±0.000 | 0.000±0.000 | 0.002±0.001 | 0.000±0.000 | 0.000±0.000 |
| n200 | 0.000±0.000 | 0.000±0.000 | 0.000±0.000 | 0.000±0.000 | 0.000±0.000 | 0.000±0.000 | 0.000±0.000 |
| n300 | 0.000±0.000 | 0.000±0.000 | 0.000±0.000 | 0.000±0.000 | 0.000±0.000 | 0.000±0.000 | 0.000±0.000 |
| Avg. | 0.000 | 0.000 | 0.033 | 0.040 | 0.026 | 0.016 | 0.001 |

influence of different grid size $W, H$, shown in Table 12. Metric 'Error' evaluates the area error between the projected region and the original region as follows:

$$\text{Error} = \frac{1}{|\mathcal{B}|} \sum_{b \in \mathcal{B}} \frac{|w\frac{W_o}{W} \times h\frac{H_o}{H} - w_o \times h_o|}{w_o \times h_o}, \tag{7}$$

where $\mathcal{B}$ is the set of all blocks. Similar performances in terms of alignment, HPWL, and overlap are achieved across different grid number settings. Lower grid numbers offer faster execution speeds but may result in relatively higher area errors that may not meet precision requirements. Higher grid numbers produce more fine-grained floorplanning results but lead to a larger action space and require additional computational resources. As a trade-off between runtime and precision, a grid size of 128 is selected.

Table 12: The influence of different grid size on circuit n100.

| Grid/Metric | Alignment | HPWL | Overlap | Runtime (s) | Error |
|---|---|---|---|---|---|
| 32 | 0.903 | 179,371 | 0.000 | 3.981 | 0.106 |
| 64 | 0.936 | 172,159 | 0.000 | 4.122 | 0.052 |
| 128 | 0.941 | 176,320 | 0.000 | 4.417 | 0.024 |
| 256 | 0.961 | 176,375 | 0.000 | 4.572 | 0.013 |
| 512 | 0.937 | 178,395 | 0.000 | 5.948 | 0.007 |

### F.5 Floorplanning with Pre-Placed Modules

Our approach is also capable of addressing the circuits with pre-placed modules (PPMs). PPMs are blocks whose position and aspect ratio are pre-determined and fixed during the FP process. Heuristics-based methods [21, 24] are incapable of address the existence of PPMs. In these heuristics approaches, after the perturbation is applied on current FP representation, the entire FP layout result may undergo changes. During this process, it cannot be guaranteed that the positions and shapes of pre-placed modules will remain unchanged. Consequently, they lack the flexibility to address circuits with PPMs. Our method FlexPlanner is capable to solve this issue. At the beginning of FP process (the beginning of an episode), all PPMs are placed on their pre-determined positions, and their aspect ratios keep unchanged. After that, we modify the canvas mask and position mask, indicating that these positions have been occupied by PPMs. In each step $t$ to place a movable block $b_t$, the overlap among $b_t$ and other blocks (PPMs and other placed blocks) can be avoided. The floorplanning result with PPMs is shown in Fig. 7.

## G Algorithms

### G.1 Algorithm for Calculation of Alignment Rate and Overlap

The calculation of alignment score $\text{aln}_t$ and overlap $o_t$ at step $t$ are shown in Alg. 3 and Alg. 4. These values are involved in the calculation of reward function with local advantage and global baseline shown in Alg. 2.

### G.2 Training Algorithm

The overall training pipeline is shown in Alg. 5, based on PPO [35] algorithm with RL framework tianshou [38]. During a training epoch, the policy network collects training data into replay buffer by interacting with the environment. To accelerate the training process, the policy network simultaneously interacts with $n_e$ environments

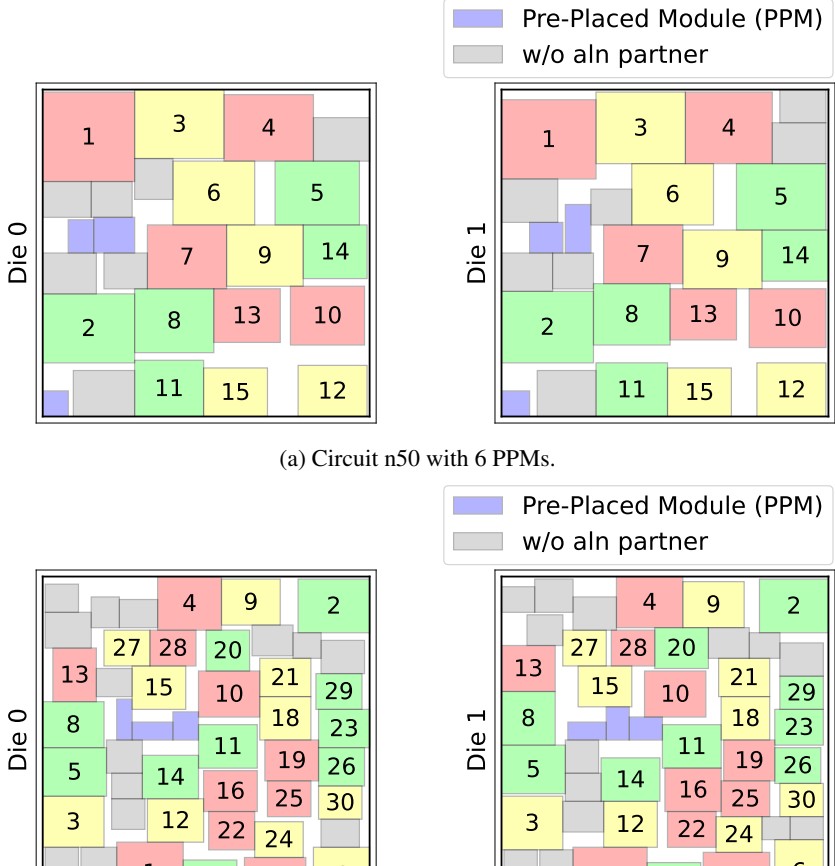

(a) Circuit n50 with 6 PPMs.

(b) Circuit n100 with 12 PPMs.

Figure 7: Our 3D floorplan result. Blue blocks are PPMs whose position and aspect ratio are fixed during FP. Two blocks with the same index on different dies form an alignment pair, which roughly locate on the same positions and share a 2D common projected area.

---

**Algorithm 3:** Alignment score calculation.

---

**Input:** Current block $b_t$, alignment score $\text{aln}_{t-1}$ at step $t-1$, number of alignment pairs $n_a$
**Output:** Alignment score $\text{aln}_t$ at step $t$
$\text{aln}_t = \text{aln}_{t-1}$
**if** $b_i$ *has alignment parnter* $b_j$ **and** *both* $b_i, b_j$ *have already been placed* **then**
$\quad \mid \quad \text{aln}_t \mathrel{+}= \frac{\text{aln}(i,j)}{n_a}$ // according to Eq. 1
**end**

---

**Algorithm 4:** Overlap calculation.

---

**Input:** Current block $b_t$, canvas mask $f_c$, chip die width $W$ and height $H$
**Output:** Overlap $o_t$ at step $t$
$f_c[z_i, x_i : x_i + w_i, y_i : y_i + h_i] \mathrel{+}= 1$
calculate overlap $o_t$ as follows:

$$o_t = \frac{\sum_{z,x,y} \max\{0, f_c[z, x, y] - 1\}}{WH}$$

---

in parallel. We set the buffer size to satisfy

$$L_{buf} \equiv 0 \bmod (n_e \times \text{len}(episode)), \tag{8}$$

ensuring that each episode in the replay buffer is completed and terminated with its final step, where $L_{buf}$ is the size of replay buffer. After data collection, we further compute the reward, cumulative discounted reward and advantage. Since each episode is completed in the replay buffer, each step has the corresponding terminated step (end step) within the same episode to calculate the reward with local advantage and global baseline. Finally, both critic network and policy network are updated.

**Algorithm 5:** Training Algorithm.

---

**for** *epoch* **from** 1 **to** *n_epochs* **do**
> // data collection
> **while** *replay buffer is not full* **do**
>> // action probability calculation
>> calculate action probability for position of current block $b_t$: $\pi_\theta\left(a_t^{(1)}|s_t\right)$
>>
>> sample $(x_t, y_t) \sim \pi_\theta\left(a_t^{(1)}|s_t\right)$
>>
>> calculate action probability for layer to next block $b_{t+1}$: $\pi_\theta\left(a_t^{(2)}|s_t\right)$
>>
>> sample $z_{t+1} \sim \pi_\theta\left(a_t^{(2)}|s_t\right)$
>>
>> calculate action probability for aspect ratio of next block $b_{t+1}$: $\pi_\theta\left(a_t^{(3)}|s_t\right)$
>>
>> sample $\mathrm{AR}_{t+1} \sim \pi_\theta\left(a_t^{(3)}|s_t\right)$
>> // action execution and next state observation
>> execute the action $a_t$:
>>> 1. place block $b_t$ at position $(x_t, y_t)$
>>>
>>> 2. access next block $b_{t+1}$ from the FIFO queue $q_{z_{t+1}}$ of layer $z_{t+1}$
>>>
>>> 3. set the aspect ratio of $b_{t+1}$ to $\mathrm{AR}_{t+1}$
>>
>> observe next state $s_{t+1}$ from environment
>> get alignment score $\mathrm{aln}_t$, wirelength $\mathrm{HPWL}_t$, overlap $o_t$ from environment
>> add sample $\left(s, s', (x, y, z, \mathrm{AR}), (\mathrm{aln}, \mathrm{HPWL}, o), (\pi_{\theta_{old}}^{(1)}, \pi_{\theta_{old}}^{(2)}, \pi_{\theta_{old}}^{(3)})\right)$ into buffer
>
> **end**
> // data processing
> **for** *each sample* **in** *buffer* **do**
>> re-compute reward $r$ with local advantage and global baseline according to Alg. 2
>> calculate state value for current state $s$: $v = V_\phi(s)$
>> calculate state value for next state $s'$: $v' = V_\phi(s')$
>> modify current sample with adding $(r, v, v')$
>
> **end**
> **for** *each sample* **in** *buffer* **do**
>> compute cumulative discounted reward $G$ and advantage $\hat{A}$ for current sample according
>> to generalized advantage estimation (GAE) [37, 35]
>> $\hat{A}_t = \sum_{i=0}^{T-t-1}(\gamma\lambda)^i \delta_{t+i}$, where $\delta_t = r_t + \gamma v_{t+1} - v_t$
>> $G_t = \hat{A}_t + v_t$
>> modify current sample with adding $(\hat{A}, G)$
>
> **end**
> // network update
> **for** *update_epoch* **from** 1 **to** *n_update_epochs* **do**
>> **for** *minibatch* **in** *buffer* **do**
>>> // policy/actor network update
>>> compute action probability $\pi_\theta^{(1)}, \pi_\theta^{(2)}, \pi_\theta^{(3)}$
>>> compute action distribution entropy $h_\theta^{(1)}, h_\theta^{(2)}, h_\theta^{(3)}$
>>> update policy/actor network according to Eq. 5
>>> // critic network update
>>> compute state value $v = V_\phi(s)$
>>> update critic network according to Sec. 3.4
>>
>> **end**
>
> **end**

**end**

---

