# OpenReview forum: "FlexPlanner: Flexible 3D Floorplanning via Deep Reinforcement Learning in Hybrid Action Space with Multi-Modality Representation"
_NeurIPS.cc/2024/Conference — NeurIPS 2024 poster_

### Official Review · Reviewer_jFwz · 2024-06-28

**Soundness:** 3
**Presentation:** 3
**Contribution:** 2
**Rating:** 5
**Confidence:** 2

**Summary:**

In this paper, the authors have proposed FlexPlanner, a flexible 3D floorplanning method with deep reinforcement learning. Existing learning methods mainly focus on the 2D scenarios. However, it suffers from overlooking alignment requirements and multi-die property. To address these, FlexPlanner learns a hybrid action space with multi-modality representation. It contains three modules to estimate position, layer and aspect ratio of blocks. Experiments demonstrate the effectiveness of the proposed method.

**Strengths:**

1. The problem it addresses is clearly stated. While most existing learning methods target at 2D FP task, this paper address 3D FP task, and also show the difficulties when directly applying 2D methods to 3D.

2. This work introduces three modalities to represent the state space, which show better representative ability than heuristics-based methods.

3. The writing of the paper is clear.

4. Significant improvements on alignment scores according to Table 2.

**Weaknesses:**

1. It would be better to include a teaser that better shows the problems when directly applying 2D FP method to this 3D scenario.

2. The novelty of the paper is a concern. From my point of view, the idea of using three modalities to address the issue is quite straightforward, which lacks of novelty. In the rebuttal, the authors should further highlight the novelty of the technical designs of this work.

3. Please further explain the significant improvements in Table 2, which is very impressive. It would be better to include some visualization comparisons as well as more explanations on why existing methods obtain such low alignment scores.

4. I have some concerns about the baselines in the table. It should include more recent baselines within 1 year. Also, I notice that [14] was published in 2010 -- why do this method still achieves comparable results with other baselines?

**Questions:**

Please see the weakness.

---

> ### Author Rebuttal · Authors · 2024-08-07
>
> ## Response to Reviewer jFwz (5: Borderline accept)
> Thank you for your time and valuable feedback. Our replies to the concerns and questions are as follows.
>
> > **W1: Better to include a teaser showing the problems when directly applying 2D FP method to 3D scenario**
>
> We sincerely appreciate your constructive suggestion. Compared to 2D FP, which involves only a single die, the 3D scenario requires placing blocks across multiple dies. The positions and shapes of the blocks on different dies should be optimized simultaneously to achieve a better result, rather than optimizing each die sequentially as in 2D FP. Furthermore, cross-die alignment becomes a critical optimization objective, and it is insufficient to rely solely on incorporating the alignment score into the heuristic cost or RL reward.
> **We will integrate corresponding discussions in the final version.**
>
> > **W2: The idea of using three modalities to address the issue is quite straightforward**
>
> We greatly appreciate your constructive concern.
> - First, **incorporating three modalities is only part of our main contributions**, in addition to other important contributions such as addressing the 3D alignment problems and zero-shot transferability, and most importantly, our approach is the first learning-based method to discard heuristic-based search in the 3D FP task and it achieves new state-of-the-art. Additionally, we propose a novel mechanism, asynchronous layer decision to determine the layer for accessing the next block to place. This approach offers greater flexibility in fine-tuning the placing order, instead of adhering to a fixed sequence.
> - Second, it is **non-trivial to directly apply multimodalities** to our approach. It should be noted that our approach is not incremental to previous works but is devised from scratch. The choices of multimodalities and other modules like alignment mask are based on large amounts of experiments and observations, and they are empirically proved to be effective in the complicated 3D FP setting with a comprehensive consideration of module position, module aspect ratio, overlap area, cross-die alignment, and the fixed outline.
> - Last, according to your suggestions, we will polish the contributions in the 'Introduction' part in our final version.
>
> > **W3: Explanation and Visualization of Improvement on Alignment Score**
>
> We provide both additional visualization comparisons and explanations on alignment score.
> - The **visualization comparisons on alignment score** are shown in Fig. 2 in the rebuttal PDF. It demonstrates that our method performs much better than other baselines on alignment score.
> - Why our method performes better
>     - **Alignment mask as representation.** The alignment mask provides RL with a more refined modeling of cross-die alignment, rather than merely perceiving alignment through reward function. This detailed guidance is crucial for the alignment task, which is inherently challenging and requires more precision.
>     - **Alignment mask as constraint.** By utilizing the alignment mask to filter out invalid positions, we place blocks in locations where the alignment constraint is satisfied. This approach reduces the action space without sacrificing quality and enhances efficiency.
>     - **Asynchronous layer decision.** This approach provides greater flexibility, allowing the method to autonomously decide the planning order, rather than completing one layer before starting the next. For instance, after placing a block on the first die, an intuitive next step is to immediately place its alignment partner on the second die to prevent other blocks from occupying the alignment region.
>
> - Why other baselines performes worse
>     - **Heuristic Methods** (e.g., 3D-B\*-SA) cannot directly constrain the action space to ensure the cross-die alignment. Instead, they calculate the alignment score and incorporate it into the heuristic cost to guide the search. While effective for smaller cases, the performance of this approach significantly deteriorates as the circuit scale increases, shown in Table 2 in main paper.
>     - **Wiremask-BBO** employs a greedy algorithm to select the highest-scoring position from the current legal positions for the next block. This greedy nature makes it prone to local optima, preventing it from achieving global optimization. Additionally, its synchronous decision-making mechanism, which completes one layer before moving to the next, is less effective.
>     - **Other RL Baselines** lack representation of the cross-die alignment, relying solely on rewards to guide optimization, which is quite limited. Ablation study in Fig. 5(a) in main text further verifies this point: removing the alignment mask either as input or as an action space constraint significantly impacts the alignment score. It demonstrates the effectiveness of our proposed alignment mask in both input representation and action space constraint.
>
> > **W4: Comparable Results of 3D-B\*-SA**
>
> We fully understand your suggestion of incorporating more recent baselines, however, wiremask-BBO is the current state-of-the-art approach and, to our best knowledge, it is the merely one within one year that can be compared due to the sophisticated experimental settings and the common closed source community of Electronic Design Automation.
>
> On the other hand, though 3D-B\*-SA was published in 2010, as a classical simulated annealing (SA) method, it is still formidable especially on small circuits, and thus, SA-based approaches are also common baselines in recent works [1,2]. However, due to the exponential increment of searching space, the performance of SA rapidly degrades with the increasement of the circuit scale and is gradually overpassed by RL in recent years.
>
> **References**
>
> [1] Generalizable floorplanner through corner block list representation and hypergraph embedding. SIGKDD. 2022.
>
> [2] Macro placement by wire-mask-guided black-box optimization. NeurIPS. 2023.

---

> > ### Comment · Reviewer_jFwz · 2024-08-09
> >
> > Thanks for your rebuttal. I would like to keep my rating of borderline accept.

---

> > > ### Author Response · Authors · 2024-08-09
> > >
> > > Thank you for your reply. Please kindly let us know if you have any follow-up questions or areas needing further clarification. Your insights are valuable to us, and we stand ready to provide any additional information that could be helpful.

---

### Official Review · Reviewer_3CLd · 2024-07-08

**Soundness:** 3
**Presentation:** 3
**Contribution:** 3
**Rating:** 7
**Confidence:** 3

**Summary:**

This paper proposes the FlexPlanner, a reinforcement learning-based method utilizing multi-modality representation, including vision, graph, and sequence, to handle different challenging scenarios.
Additionally, the design of the action space to uniformly handle constraints represents a new and innovative method.
This approach, being largely free from heuristics, is more aligned with machine learning principles and advances the field of learning for floorplanning, particularly in 3D scenarios.
The performance improvement from 0.474 to 0.940 on public benchmarks is significant, and the method also demonstrates strong transfer learning capabilities.

**Strengths:**

1. This paper is well written, with a clear background and motivation, and Table 1 is informative and categorizes the features of existing methods, which helps me understand the approach.

2. The use of vision, graph, and sequence modalities provides a rich representation of the planning problem, expanding the action space and enabling the learning of multiple properties, such as position, aspect ratio, and layer for each block.

3. The experimental results are much better than previous methods on benchmarks in terms of wirelength and alignment.

4. The appendix part is also informative and useful for better understanding the approach and the results

**Weaknesses:**

There is no weakness in this paper, to be honest.

**Questions:**

As RL could be a controversial approach for placement and routing, how do the authors defend their RL-based methodology to the floorplanning problem?

**Limitations:**

No specific limitations.

---

> ### Author Rebuttal · Authors · 2024-08-07
>
> ## Response to Reviewer 3CLd (7: Accept)
> Thank you for your time and valuable feedback. Our replies to the questions are as follows.
>
> > **Q1: Defend RL on Floorplanning Task**
>
> Thanks for your insightful comment. Compared to other methods, our RL-based approach offers the following advantages in 3D floorplan task:
>
> - **Better to handle hard constraints compared to analytical methods**: Analytical methods often require smooth approximations of optimization objectives to make them differentiable, which reduces precision and accuracy. Moreover, many optimization objectives, such as alignment score, are difficult to approximate. As a result, it is hard for analytical algorithms to optimize these objectives. In contrast, RL models optimization objectives through reward designs, where differentiability of reward calculation is not a prerequisite. As a consequence, approximations are unnecessary, resulting in greater precision. Additionally, analytical methods struggle to meet hard constraints (e.g., cross-die module alignment constraint, non-overlap constraint), typically relaxing them to soft constraints via penalty terms. On the contrary, our FlexPlanner introduces masks (e.g., alignment mask and position mask) to directly handle hard constraints in the action space by filtering out invalid positions.
>
> - **Higher performance upper bound and more flexibility compared to heuristic algorithms**: The solution space modeled by heuristics representation is limited and cannot encompass all solutions, making it challenging to find the optimal solution. Besides, they lack flexibility, making it difficult to precisely fine-tune the position of each block. On the contrary, our RL approach, FlexPlanner, leverages a hybrid action space with more flexibility. It directly determines the coordinate, aspect ratio and layer for each block. Moreover, Heuristic methods can only guide the searching process through cost functions and fail to satisfy specific hard constraints. In FlexPlanner, masks are incorporated to enforce hard constraints in the action space.
>
> - **Generalization performance**: Our method benefits from a general multimodal representation and a unified reward function design, enabling fine-tuning and zero-shot inference capabilities across different circuits. Moreover, our method employs **the same hyperparameter settings for all circuits**, demonstrating its robustness and generalizability.
>
> **We will integrate corresponding discussions in the final version.**

---

> > ### Comment · Reviewer_3CLd · 2024-08-13
> > **feedback**
> >
> > Thank you for your response.My questions are well addressed,  I maintain my score.

---

> > > ### Author Response · Authors · 2024-08-13
> > >
> > > Thank you for your reply. We will integrate corresponding discussions in the final version.

---

### Official Review · Reviewer_pH9C · 2024-07-09

**Soundness:** 3
**Presentation:** 2
**Contribution:** 3
**Rating:** 5
**Confidence:** 2

**Summary:**

This paper presents a new learning-based method for IC design that simultaneously handles the position, aspect ratios, and alignment of blocks.  The method achieves significant improvements compared to baselines by leveraging reinforcement learning with a hybrid action space and multi-modality representation to optimize block positions, aspect ratios, and cross-die alignments.

**Strengths:**

* The application of RL with a novel reward function in a hybrid action space to 3D IC FP is very interesting and novel, moving away from traditional heuristic-based methods.
* The paper demonstrates notable improvements compared to existing baselines on the evaluated benchmarks.
* The paper clearly defines the used metrics.

**Weaknesses:**

Clarity and Writing:
* While the paper is overall well-organized, some sections could benefit from additional explanations or simplifications to make the content more accessible to readers without an IC design background.
* Important design details  (e.g., the critic network) are deferred to the appendix and not introduced in the main text.
* Although the benchmarks and baselines used are relevant, discussing them in more detail could significantly enhance the presentation.


Evaluation Pipeline:
* Providing more details on the exact setup of training and test data could improve transparency.
* RL methods are known to suffer from stability issues and are often sensitive to hyperparams. How sensitive is the method to the hyperparams used? This should be evaluated and discussed to comment on the reproducibility of the results.
* How well does the method generalize to larger circuits beyond those in the benchmarks? A discussion in this direction would be highly appreciated.

Design Choices:
* There are many design choices in the paper. For instance, the significance of the multi-modality input is not entirely clear. The model takes in "vision," "graph," and "sequence" modalities, but it is not specified how much each of these contributes to the final performance. While the ablation studies show that the alignment mask is a good design choice, they do not provide much intuition about the importance of other design choices, such as the canvas mask or input modalities (e.g., sequence and graph). Are these necessary?

**Questions:**

see above.

**Limitations:**

Yes, the limitations are briefly addressed in the conclusion.

---

> ### Author Rebuttal · Authors · 2024-08-07
>
> ## Response to Reviewer pH9C (5: Borderline accept)
> Thank you for your time and valuable feedback. Our replies to the concerns and questions are as follows.
> > **W1: Clarity and Writing.**
>
> According to your suggestions, we have polished the paper in the following aspects:
> - We will provide **more explanations for readers without an IC background**. For example, in Fig.2 in rebuttal PDF, we provide more vivid visualizations of cross-die alignment to help readers understand its definition and significance.
> - We will revise the paper to include **key design details** (e.g., critic network), ensuring that readers have a comprehensive understanding without refering to appendix.
> - In the final version, we will give more detailed analysis and discussion on **benchmarks and baselines.**
>
> **All revisions will be updated in the final version.**
>
> > **W2.1: More details about training and test data**
>
> - For **experiments of training from scratch**, RL is trained **case by case** by PPO via interaction with floorplan environment. After training, we evaluate its performance on this circuit.
> - For **experiments of fine-tuning and zero-shot inference**, we train RL on circuit n100 and test it on other circuits.
>
> We will clarify the above details in the final version.
>
> > **W2.2: Sensitivity/Ablation on Hyperparams**
>
> According to your suggestions, we make the following supplements:
> - **Additional experiments to evaluate the sensitivity** to hyperparameters, such as learning rate, mini-batch size. Experiments are shown in Figure 1 in the rebuttal PDF. Stable training curves show that our approach achieves good stability on different hyperparams settings.
> - Moreover, our method is capable of employing the **same hyperparameter settings for all circuits**, rather than adjusting hyperparameters for each circuit. It shows the robustness and generalizability of our method.
>
> > **W2.3: Generalization on Larger Circuits**
>
> We sincerely appreciate your suggestion. MCNC and GSRC benchmarks are indeed widely used datasets for 2D and 3D FP tasks. The largest case in these benchmarks is the circuit n300, with 300 blocks, 569 I/O ports, and 1893 nets. It is sufficiently large, even by industry standards. Recently, Intel, a leading CPU manufacturer, released FloorSet [1] (accepted by ICCAD 2024 but not published), a VLSI floorplanning dataset derived from real-world System on Chip (SoC). In FloorSet, the largest case contains 120 blocks, which is **significantly smaller** than n300 in our evaluation pipeline. Given that FloorSet is based on actual SoCs, we believe that current scale of our benchmarks is adequate to ensure robust performance in **real-world industrial applications**.
>
> To further demonstrate the capability of our approach, we design additional experiments on larger circuits shown in Table 2 in rebuttal pdf (or see below).
> |Circuit|adaptec2|adaptec2|adaptec2|adaptec3|adaptec3| adaptec3 | n300_dup3 | n300_dup3 | n300_dup3 |
> |:-:|:-:|:-:|:-:|:-:|:-:|:-:|:-:|:-:|:-:|
> ||#Block|#Net|#I/O Port|#Block|#Net|#I/O Port|#Block|#Net| #I/O Port|
> ||566|860|0|723|1,154|0|900|5,679|1,707|
> |Method/Metric|Alignment|HPWL|Overlap|Alignment|HPWL|Overlap|Alignment|HPWL|Overlap|
> |Wiremask-BBO|0.285|2,956,546|0.000|0.365|5,000,467 |0.000|0.391|2,389,902|0.000|
> |Ours|**0.839**|**2,911,438**|**0.000**|**0.817**| **4,758,283**|**0.000**|**0.928**|**2,305,070**|**0.000**|
>
> **1. Training from scratch**
> ISPD 2005 benchmark [2] is a standard for 2D global placement (GP), a task performed after floorplanning. In this benchmark, each circuit consists of more than 500 macros (large functional blocks) and millions of cells (tiny logic gates). The primary objective of GP is to optimize locations of cells to minimize wire length.
> For our purposes, we remove cells from the chip canvas and netlist, constructing new circuits (with more blocks than n300) that consists solely of macros. Macros are then assigned to two dies, and alignment pairs are constructed. We evaluate our method on these modified circuits. Experiments show that our approach achieves an alignment score with 0.839 and 0.817, significantly surpassing 0.285 and 0.365 by Wiremask-BBO.
>
> **2. Zero-shot inference**
> We construct a synthetic circuit, n300_dup3, by duplicating all components of circuit n300 three times. It results in a larger circuit with 900 blocks, 1707 I/O ports, and 5679 nets.
> For this circuit, we perform inference directly using the pre-trained checkpoint obtained from training on circuit n100. It achieves an alignment score of 0.928, significantly surpassing 0.391 by Wiremask-BBO. It shows the capability of zero-shot inference, even when the testing circuit is nine times larger than the training circuit.
>
> **References**
>
> [1] FloorSet-a VLSI Floorplanning Dataset with Design Constraints of Real-World SoCs. ICCAD. 2024.
>
> [2] The ISPD2005 placement contest and benchmark suite. ISPD. 2005.
>
> > **W3: Ablation on Design Choices**
>
> Thank you for your suggestions. Actually, in Table 5 in main paper, we have already compared our FlexPlanner **without sequence (w/o seq)** or **without graph (w/o graph)**. They are inferior to FlexPlanner with full modalities, so the choices of graph and sequence are necessary.
>
> For more sufficient and clearer comparisons, we conducte **further ablation studies** in Table 1 in rebuttal PDF (or see below). Vision includes alignment mask, wire mask, position mask and canvas mask. Specifically, we compare more combinations of three modalities. As shown in the Table 1, vision is the most important but graph and sequence can also promote the effectiveness.
> ||Method|MaskPlace|Ours|GraphPlace|DeepPlace|Ours|Ours|Ours |Ours|Ours|
> |:-:|:-:|:-:|:-:|:-:|:-:|:-:|:-:|:-:|:-:|:-:|
> |Modality|Vision|✔|✔|✔|✔|✔||✔|✔(no canvas mask)|✔|
> ||Graph|||✔|✔|✔|✔ ||✔|✔|
> ||Sequence||||||✔|✔|✔|✔|
> |Metric|Alignment|0.575|**0.847**|0.279|0.332|**0.860**|0.301|0.874|0.744|**0.961**|
> ||HPWL|189,100|**187,026**|209,940|223,359|**186,005**|221,602|185,079|186,732|**176,639**|

---

### Official Review · Reviewer_5TLE · 2024-07-15

**Soundness:** 3
**Presentation:** 3
**Contribution:** 3
**Rating:** 5
**Confidence:** 3

**Summary:**

The paper proposes a learning-based method called FlexPlanner  in hybrid action space with multi-modality representation to simultaneously handle position, aspect ratio, and alignment of blocks.  FlexPlanner models 3D FP based on multi-modalities, includ15 ing vision, graph, and sequence.  The work designs a policy network with hybrid action space and asynchronous layer decision mechanisms that allow for determining the versatile properties of each block.  Experiments on public benchmarks MCNC and GSRC  show the effectiveness.

**Strengths:**

1. The paper introduces a new learning-based method in hybrid action space with multi-modality representation for 3D floorplanning task.
2. The paper explains the rationale behind each proposed module/component of the model with experimental verifications of the module's effectiveness.

**Weaknesses:**

Lack of Clarity: Certain sentences in the paper are unclear and difficult to understand, hindering the comprehension of the proposed methodology.  For example, the elaborated form of ablation experiment section is similar to an experiment report.

Insufficient Experimental: The paper may lack some ablation experiments about the important hyperparameters.

**Questions:**

Unfair comparisons: There are unfair comparisons with other work. For instance, the methods  GraphPlace [35] and DeepPlace [13] originally incorporated graph and vision as representations.  MaskPlace [25] only employed visual representation, whereas the authors’ method used graph, vision and sequence. Three multimodalities provide more information.

**Limitations:**

Yes

---

> ### Author Rebuttal · Authors · 2024-08-07
>
> ## Response to Reviewer 5TLE (5: Borderline accept)
> Thank you for your time and valuable feedback. Our replies to the concerns and questions are as follows.
>
> > **W1: Clarity**
>
> In the final version, we will conduct more ablations studies in terms of different hyperparameters and the impact of different modalities, and integrate more detailed corresponding introduction. Additionally, we further polish the paper and clarify the following aspects:
>
> - We refresh the ablations study table and add more analysis including trying different combinations of modalities in Table 1 in the rebuttal PDF.
>
> - Additionally, we detail the introduction, analysis, and experimental protocals, and **polish the content** in our final version, including:
>
>   - **Additional Explanations on the Background of Integrated Circuit (IC) Design.** We add more background about ICs and introduce the vivid visualization of cross-die alignment, as shown in Fig. 2 in our rebuttal PDF, to assist readers to understand its definition and significance.
>   - **Design Details.** We will revise the paper to include critical design details, such as the critic network, ensuring that readers have a comprehensive understanding without needing to refer to the appendix.
>   - **Details of Benchmarks and Baselines.** In the final version, we will expand this section to provide a thorough analysis, highlighting the relevance and implications of our choices.
>
>
> > **W2: Ablation on Hyperparameters**
>
> We appreciate your valuable comments. According to your suggestions, we have made the following supplements:
>
> - We provide **additional experiments to evaluate the sensitivity** of RL with respect to hyperparameters, such as learning rate, mini-batch size. Experiments are shown in Figure 1 in the rebuttal pdf. It demonstrates that our approach achieves good stability on different hyperparams settings (stable training curve across different hyperparameters).
> - Moreover, our method is capable of employing the **same hyperparameter settings for all circuits**, rather than adjusting hyperparameters individually for each circuit. It demonstrates the robustness, generalizability of our method.
>
>
> > **Q1: Comparisons With Same Multimodalities**
>
> We greatly appreciate your constructive concern.
> - First, we believe that **incorporating multimodalities is one of our main contributions** (the second contribution in 'Introduction' part) and to our best knowledge, it is the first trial to simultaneously incorporate vision, graph, and sequence.
> - Second, it is **non-trivial to directly apply multimodalities** to existing baselines considering the model convergence and the more complicated setting in our paper.
> - Last, we fully acknowledge and understand the latent unfairness. To address this, in addition to the existing ablation study in Table 5, we conducted **further ablation studies**, as presented in Table 1 in the rebuttal PDF (or see below for clarification). Specifically, we display the alignment and HPWL for our FlexPlanner using the same modalities as DeepPlace, GraphPlace, and MaskPlace. As shown in the table, FlexPlanner surpasses all other baselines in both alignment and HPWL with the same modalities.
>
> | |Method|MaskPlace|Ours|GraphPlace|DeepPlace|Ours|Ours|Ours |Ours|Ours|
> |:-:|:-:|:-:|:-:|:-:|:-:|:-:|:-:|:-:|:-:|:-:|
> |Modality| Vision | ✔ | ✔ | ✔ | ✔ | ✔ | | ✔ |✔ (no canvas mask) |✔ |
> | | Graph | | | ✔ | ✔ | ✔ |✔ | |✔ |✔ |
> | | Sequence| | | | | |✔ |✔ |✔ |✔ |
> | Metric |Alignment| 0.575 |**0.847** | 0.279 | 0.332 |**0.860** |0.301 |0.874 | 0.744 | **0.961** |
> | | HPWL | 189,100 |**187,026**| 209,940 | 223,359 |**186,005**|221,602|185,079|186,732 |**176,639**|

---

### Author Rebuttal · Authors · 2024-08-07

## Global Response
Dear Area Chairs and Reviewers,

We appreciate your time, valuable comments, and constructive suggestions. From an overall perspective, we are happy to see that **all reviews are positive** and the reviewers approve of the **novelty** (`3CLd`, `pH9C`, `5TLE`), **notable improvements** (`jFwz`, `3CLd`, `pH9C`), and **strong transfer learning capabilities** (`3CLd`). Additionally, we are grateful for the acknowledgment that this paper is **well-motivated** (`jFwz`,  `3CLd`) and **well-organized** (`pH9C`, `jFwz`, `3CLd`).

According to the reviewers' suggestions, we have made the following revisions:

- **More Ablation Studies.** In addition to the existing ablation study in Table 5 in main text, we conduct further ablation studies, as presented in Table 1 in the rebuttal PDF (or see below for clarification). Specifically, we display the alignment and HPWL for our FlexPlanner using the same modalities as DeepPlace, GraphPlace, and MaskPlace. As shown in the table, FlexPlanner surpasses all other baselines in both alignment and HPWL with the same modalities.

| | Method |MaskPlace| Ours |GraphPlace|DeepPlace| Ours | Ours |Ours |Ours |Ours |
|:-:|:-:|:-:|:-:|:-:|:-:|:-:|:-:|:-:|:-:|:-:|
|Modality| Vision | ✔ | ✔ | ✔ | ✔ | ✔ | | ✔ |✔ (no canvas mask) |✔ |
| | Graph | | | ✔ | ✔ | ✔ |✔ | |✔ |✔ |
| | Sequence| | | | | |✔ |✔ |✔ |✔ |
| Metric |Alignment| 0.575 |**0.847** | 0.279 | 0.332 |**0.860** |0.301 |0.874 | 0.744 | **0.961** |
| | HPWL | 189,100 |**187,026**| 209,940 | 223,359 |**186,005**|221,602|185,079|186,732 |**176,639**|

- **More Clarity.** We detail the introduction, analysis, and experimental protocals, and polish the content in our final version, including:
    - **Additional Explanations on the background of Integrated Circuit (IC) design.** We add more background about ICs and introduce the vivid visualization of cross-die alignment, as shown in our rebuttal pdf, to assist readers to understand the definition and significance of it.
    - **Design Details.** We will revise the paper to include critical design details, such as the critic network, ensuring that readers have a comprehensive understanding without needing to refer to the appendix.
    - **Details of Benchmarks and Baselines.** In the final version, we will expand this section to provide a thorough analysis, highlighting the relevance and implications of our choices.

**All the above revisions will be updated in the final version.**

**A one-page PDF is uploaded that contains corresponding tables and figures in the response.**

In the following, we provide detailed answers. We are glad to further response for informed evaluation.

---

### Author Response · Authors · 2024-08-12
**To Reviewers: Kindly Request Your Constructive Feedback as the Rebuttal Deadline Approaches**

Dear Reviewers and ACs,

We are grateful to all the reviewers and area chairs for your efforts and valuable feedback. We understand that the reviewing process is very demanding and time-consuming, and we respect the reviewers' expertise and capability. However, as the rebuttal deadline is approaching, we believe sufficient discussion and communication between the authors and reviewers are important for making the final informed and conclusive decision. We kindly request your valuable comments on our rebuttal and your attention to our clarifications.

Thank you for your time and consideration.

Sincerely,
Paper 8140 Authors

---

### Decision · Program_Chairs · 2024-09-25

**Decision:**

Accept (poster)

**Comment:**

The submitted paper proposed a method for "floor planning", ie determining the positions and shapes of blocks in integrated circuit design, with the difficulty being the emergence of 3D circuits. Five expert reviewers have provided a favorable assessment. They appreciated novelty (an RL formulation with a hybrid action space, based on Hybrid PPO), motivation, performance gains,

Some weaknesses were raised, the paper somewhat lacking in clarity in some aspects, doubts on comparisons with prior work, information on stability in RL training, generalization, abundane of design choices.

A consensus was reached that the paper is a worthy contribution for the NeurIPS community, with the strengths
The AC concurs.